

# Calibration of radar differential reflectivity using quasi-vertical profiles

Daniel Sanchez-Rivas and Miguel A Rico-Ramirez

Department of Civil Engineering, University of Bristol, Bristol, BS8 1TR, United Kingdom

**Correspondence:** Daniel Sanchez-Rivas (d.sanchezrivas@bristol.ac.uk)

**Abstract.** The differential reflectivity ($Z_{DR}$) is a crucial weather radar measurement that helps to improve quantitative precipitation estimates using polarimetric weather radars. However, a system bias between the horizontal and vertical channels generated by the radar produces an offset in $Z_{DR}$. Existing methods to calibrate $Z_{DR}$ measurements rely on vertical observations of $Z_{DR}$ taken in rain, in which $Z_{DR}$ values close to 0 dB are expected. However, not all weather radar systems are capable

of producing vertical pointing measurements. In this work, we present and analyse a novel method for correcting and monitoring the $Z_{DR}$ offset using quasi-vertical profiles of polarimetric variables. The method is applied to radar data collected through one year of precipitation events by two operational C-band weather radars in the UK. The proposed method proves effective in achieving the required accuracy of 0.1 dB for the calibration of $Z_{DR}$ as the calibration results are consistent with the traditional method based on vertical profiles. Additionally, the method is independently evaluated using disdrometers located near

the radar sites. The results showed a good agreement between disdrometer-derived and radar-calibrated $Z_{DR}$ measurements.

## 1 Introduction

The differential reflectivity ($Z_{DR}$) plays a key role in quantitative precipitation estimation (QPE) algorithms using polarimetric weather radars. Its relation with the orientation, shape and size of the hydrometeors improves the accuracy of radar QPE algorithms and hydrometeor classification systems (Bringi et al., 2011; Chandrasekar and Bringi, 1988; Cifelli et al., 2011;

Giangrande and Ryzhkov, 2008; Gou et al., 2019; Al-Sakka et al., 2013; Park et al., 2009; Ryzhkov et al., 2005b; Vulpiani et al., 2009).

But to incorporate $Z_{DR}$ as a valid input for QPE, it is necessary to ensure that it is properly calibrated. Ryzhkov et al. (2005a) showed that an accuracy of 0.2 dB in the differential reflectivity calibration is desirable for practical applications of polarimetric weather radar data, as this figure generates uncertainty in the rain estimates close to 18%. However, several factors

introduce a bias in $Z_{DR}$, e.g. (a) the presence of cross-polar radiation (Zrnić et al., 2010); (b) errors in the transmitter and/or receiver chain (Zrnic et al., 2006) or (c) an overall system bias due to the ratio of power transmitted to the horizontal and vertical polarisations (Bringi and Chandrasekar, 2001).

To correct the overall system bias (or offset) in $Z_{DR}$, several calibration procedures have been proposed depending on the radar scanning strategy. For radars capable of performing measurements at 90° elevation angle (herein referred to as birdbath

scans), the most accepted calibration procedure is based on the observation of raindrops measured in light rain as the radar



antenna rotates about the vertical; non-zero values of $Z_{DR}$ present under these conditions can be set as the $Z_{DR}$ offset. This method was introduced by Gorgucci et al. (1999) and has been further explored and validated on several radar campaigns, e.g., Bechini et al. (2002) used vertical profiles (VPs) generated from data collected by a weather radar located in Italy to estimate both the $Z_{DR}$ offset and the error in the radar reflectivity ($Z_H$). Similarly, Gourley et al. (2009) estimated the $Z_{DR}$ offset using

birdbath scans collected by a C-band radar and demonstrates its impact on the absolute calibration of $Z_H$; Louf et al. (2019) used polarimetric birdbath scans measured by a C-band radar located in Australia to validate a new approach for calibrating and monitoring $Z_H$ using ground clutter and satellite data. Frech and Hubbert (2020) used data collected from the radar network operated by the German Meteorological Service to monitor the $Z_{DR}$ calibration. Their method relies on range-averaged values of $Z_{DR}$ collected in light rain and detected using thresholds on polarimetric variables like the co-polar correlation coefficient

or the coherent power to target an accuracy on $Z_{DR}$ of around $\pm 0.1$ dB. More recently, Ferrone and Berne (2021) expanded the birdbath method by estimating the offset based on interpolated $Z_{DR}$ values taken from rain, snow, or ice regions, being the main advantage of this method its applicability when rain regions are not available to estimate the $Z_{DR}$ offset.

However, some weather radar networks are unable to perform birdbath scans due to mechanical constraints. So several procedures have been proposed to overcome this restriction and correct the $Z_{DR}$ offset. Ryzhkov et al. (2005a) presented a

method based on the $Z_{DR}$ values of dry aggregates collected at elevation angles between $40°$ and $60°$. They linked these values to the $Z_{DR}$ offset, achieving an accuracy of 0.2 dB. Giangrande and Ryzhkov (2005) expanded this method for scans affected by the presence of partial beam blockage and explored its relation with the $Z_{DR}$ offset, stating that this method achieves an accuracy of 0.3 dB when applied to large data sets. It is worth noting that both methods require a pre-classification of the hydrometeors before their implementation. Bechini et al. (2008) proposed a method to quantify the $Z_{DR}$ offset by probing the

differential reflectivity while increasing the elevation angles but below the melting layer (ML). Then, this data is compared with theoretical profiles of $Z_{DR}$ to estimate the $Z_{DR}$ offset. Although it is possible to achieve high accuracy by applying this method ($\sim 0.1$ dB), thousands of profiles are needed to generate profiles suitable for the comparison process.

Another well-known technique to calibrate $Z_{DR}$ relies on sun measurements. It is based on the detection of solar spike echoes as this type of radiation has equal power at both horizontal and vertical polarisations (Gourley et al., 2006), hence generating

measurements of $Z_{DR}$ close to 0 dB. The sun-radiation detection method has been further investigated in several works, e.g. Gourley et al. (2006) compared both the birdbath scans and sun radiation detection methods using C-band polarimetric data, determining that higher accuracy is achieved when using the former. An on-line variation of the solar-radiation detection method that does not require the operational scanning strategy to be stopped, was introduced by Holleman et al. (2010). It is based on other works conceived to monitor the absolute radar calibration, like the methods introduced by Darlington et al.

(2003) and Huuskonen and Holleman (2007). This on-line method enables monitoring the calibration of $Z_{DR}$ and also the analysis of the correlation between horizontal and vertical lobes. Later, Huuskonen et al. (2016) expanded this method based on data collected from the Finnish radar network, adding quality control to the solar signals and achieving accuracy on $Z_{DR}$ below 0.05 dB. Chu et al. (2019) also used the sun radiation detection method and concluded that an accurate calibration depends on the availability of radar data taken at sunrise/sunset, among other considerations. It is worth noting that the offset





detected by the solar method must be taken with care as it is related to the receiver chain only, whereas the offset computed from birdbath scans includes both the transmitter and the receiver chain (Huuskonen et al., 2016).

Some other alternative techniques have been proposed to complement the operational calibration and monitoring of $Z_{DR}$. Bringi et al. (2006) estimated the $Z_{DR}$ offset of a C-band polarimetric radar located in Japan by fitting intrinsic ice values of $Z_{DR}$ to the mean values of a Gaussian distribution, whereas Richardson et al. (2017) proposed the use of turbulent eddies to

monitor the differential reflectivity as the nature of such scatters results in values of $Z_{DR}$ close to 0 dB. Additionally, Ryzhkov et al. (2016) proposed the application of the quasi-vertical profiles (QVPs) approach to monitor the calibration of $Z_{DR}$ using a similar rationale as in Ryzhkov et al. (2005a). This approach is explored by Griffin et al. (2020) or Kumjian et al. (2016), in which previously offset-corrected QVPs of $Z_{DR}$ are used to describe processes like the ML and ice aggregation/riming. Although the QVPs are a useful tool for monitoring the temporal evolution of precipitation and certain microphysics of pre-

cipitation, there is little to none amount of research using QVPS in rain to estimate the $Z_{DR}$ offset.

This study presents an operational method to correct the $Z_{DR}$ offset that can be implemented using quasi-vertical profiles of polarimetric variables. The method is based on QVPs generated from elevation angle scans of around 10° and collected during light rain. These scans are usually available in operational radar scanning strategies deployed in radar networks worldwide, thus becoming a great option for radar networks not capable of collecting measurements at vertical incidence. The C-band

polarimetric weather radars developed by the UK Met Office (UKMO) can perform measurements at vertical incidence, allowing a thorough comparison of the performance of both methods. Additionally, we explore the temporal variation of the $Z_{DR}$ offset using long-term observations collected by two operational weather radars. The calibrated $Z_{DR}$ measurements are further compared with measurements from independent disdrometer observations located near the radar sites. The paper is organised as follows. In Section 2, we define the radar and disdrometer data sets used in this work. The two different methods used to

calibrate the radar $Z_{DR}$ are described in Section 3. In Section 4, we examine the performance of the proposed method using long-term data sets collected by two weather radars and several disdrometers. We discuss the methods and results in Section 5. Finally, we summarise the findings of this work in Section 6.

## 2 Data sets

### 2.1 Radar data sets

The raw polarimetric radar data sets were obtained from two C-band weather radars part of the UKMO operational weather radar network. The Chenies radar site is located at Hertfordshire, near London, United Kingdom (Met Office, 2013) and the Dean Hill radar site is located at Wiltshire, near Salisbury, United Kingdom (Met Office, 2021). Both radars transmit and receive horizontal and vertical polarisations simultaneously, generating Plan-Position-Indicator (PPI) products sampled using different length pulses and covering several elevation angles. The PPI products containing measurements of reflectivity ($Z_H$),

differential reflectivity ($Z_{DR}$), correlation coefficient ($\rho_{HV}$), differential propagation phase ($\Phi_{DP}$) and radial velocity ($V$) were collected throughout 2018 to carry out a long-term analysis of the $Z_{DR}$ calibration; such products and its processing are described next.





The birdbath scans are sampled at vertical incidence with a temporal resolution of 10 minutes, 75 metres of gate resolution and a maximum range cover of 12 kilometres. These products are used to build VPs of polarimetric variables and monitor
the $Z_{DR}$ calibration. The VPs are generated by averaging azimuthally raw polarimetric data taken from birdbath scans but avoiding the first kilometre in height to minimise the risk of side-lobes contamination and the presence of other artefacts that could affect the $Z_{DR}$ calibration. Also, the VPs are used as input for a ML detection algorithm to distinguish the precipitation in the liquid phase, as described in Sanchez-Rivas and Rico-Ramirez (2021).

PPI scans at 9° elevation angle are collected every 10 minutes and sampled on Short Pulse (SP) mode (pulse length equal
to 500 $\mu s$), with a gate resolution of 600 metres and a maximum range of 115 kilometres. These scans are processed to generate QVPs of polarimetric variables following the procedure suggested by Ryzhkov et al. (2016), averaging azimuthally the polarimetric variables and generating one QVP of each polarimetric variable per PPI scan. As above, these data are also used to detect the ML. The QVPs will also be used to calibrate and monitor $Z_{DR}$.

PPI scans at 0.5°, 1.0° and 2.0° elevation angles are collected every 5 minutes and sampled on Long Pulse (LP) mode (pulse
length equal to 2,000 $\mu s$), covering a range of 250 kilometres and same gate-resolution as above. These low-elevation angles are used to compare calibrated radar $Z_{DR}$ with $Z_{DR}$ values derived from disdrometer data. To remove non-meteorological echoes, a fuzzy logic classifier is applied using the methodology proposed by Rico-Ramirez and Cluckie (2008). Once the differential reflectivity has been calibrated, corrections for attenuation in $Z_H$ and $Z_{DR}$ are applied following the methods presented by Rico-Ramirez (2012) and Bringi et al. (2001) respectively.
The location and other relevant technical details of the radars are provided in Figure 1 and in Tables 1 and 2.

**Table 1.** Radars characteristics.

| Description | Wave length | Scanning strategy | Beam width | PRF | RPM |
|---|---|---|---|---|---|
| Chenies & Dean Hill C-band weather radars | 5.3 cm | 8 elevations (0.5°, 1°, 2°, 3°, 4°, 6°, 9° and 90°) | 1.0° | 900 $Hz$ (SP) - 300 $Hz$ (LP) | 3.6 (SP) - 1.4 (LP) |

*Note:* PRF=Pulse Repetition Frequency; RPM=Revolutions Per Minute.

## 2.2 Disdrometer data sets

In this study, disdrometer data are used for verifying the consistency of the radar differential reflectivity measurements as the disdrometers are instruments that measure the drop size distribution (DSD) of precipitation. Several disdrometer data sets were collected from different projects with neighbouring locations to the radar sites and matching time periods. These
include disdrometers from the Chilbolton Facility for Atmospheric and Radio Research (CFARR), the Disdrometer Verification Network (DiVeN) and the University of Bristol (UoB) (see locations in Figure 1 and Table 2).

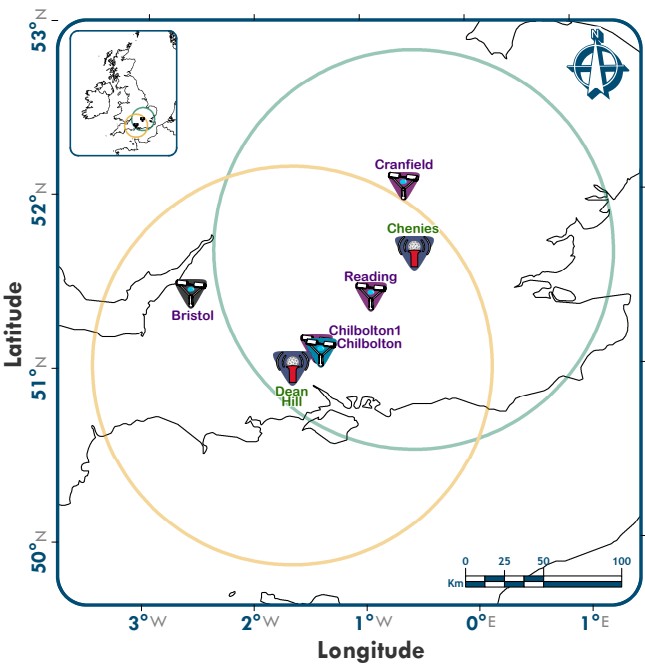

**Figure 1.** Location of the radars (Chenies and Dean Hill) and disdrometers (Bristol, Chilbolton, Reading and Cranfield). The circles represent the coverage of each radar at a distance of 115 km (maximum coverage of the radars operating at short pulse).

CFARR operates a Joss-Waldvogel impact disdrometer (model RD-69) located at Chilbolton, Hampshire, south England, providing continuous DSD data since 2003. The disdrometer converts the vertical momentum of a falling drop into signals whose amplitude depends on the diameter of the impacting drop. This device provides drop counts every 10 seconds over 127

bins ranging from 0.3 to 5 mm, with a sampling area of approximately 50 $cm^2$ (Science and Technology Facilities Council et al., 2003). This instrument does not measure the fall velocity of precipitation particles, and therefore the device does not provide a hydrometeor classification. For this work, data were available from January 2018 to July 2018.

DiVeN was deployed in 2017, installing several Thies laser precipitation monitors in the UK and providing information on the quantity, intensity and type of precipitation (Pickering et al., 2019). The Thies disdrometers classify the hydrometeors

into one of 20 diameter bins from 0.125 mm to 8 mm, and one of the 22-speed bins from 0.0 $m\,s^{-1}$ to 20.0 $m\,s^{-1}$ (Natural Environment Research Council et al., 2019). The sampling area of this instrument is approximately 45.6 $cm^2$. For this work, we selected three disdrometers operating near the radar sites, one at Chilbolton, Hampshire (herein Chilbolton1), one at Reading, Berkshire and one at Cranfield, Bedfordshire. Data were collected for precipitation events throughout 2018.

The UoB operates several Parsivel2 disdrometers, one of them located at Bristol, southwest England. This disdrometer

categorises the precipitation particles in several classes, based on the volume equivalent diameter (32 bins ranging from 0 to 25 mm) and the particle speed (32 bins ranging from 0.2 to 20 $m\,s^{-1}$), providing reliable precipitation data including hydrometeor





type and precipitation intensity (OTT HydroMet, 2016). The sampling area of this instrument is approximately 50 $cm^2$. Data were collected for precipitation events throughout 2018.

**Table 2.** Radars (RAD) and disdrometers (DIS) locations and types.

| Site Name | Facility | Model | D-CH | D-DH |
|---|---|---|---|---|
| Chenies | UKMO (RAD) | In-house design | - | 107.18 km |
| Dean Hill | UKMO (RAD) | In-house design | 107.18 km | - |
| Chilbolton | CFARR (DIS) | Joss-Waldvogel RD-69 | 87.40 km | 19.79 km |
| Chilbolton1 | DiVeN (DIS) | Thies | 87.40 km | 19.79 km |
| Cranfield | DiVeN (DIS) | Thies | 43.28 km | 136.28 km |
| Reading | DiVeN (DIS) | Thies | 39.39 km | 67.81km |
| Bristol | UoB (DIS) | Parsivel | 143.66 km | 76.19 km |

*Note:* D-CH=Distance to the Chenies radar site; D-DH=Distance to the Dean Hill radar site.

### 2.2.1 Processing of disdrometer data.

The raindrop size distribution (DSD) can be computed from the disdrometer data by Ji et al. (2019):

$$N_m(D_i) = \frac{n_i(t)}{A.\Delta t.V_i.\Delta D_i} \tag{1}$$

where $D_i$ is the drop diameter, $n_i$ is the number of drops counted during the sampling interval $\Delta t$ at the $i$th bin size, $A$ ($m^2$) is the sampling area of the disdrometer, $V_i$ ($m\,s^{-1}$) is the terminal velocity of the raindrops at the $i$th bin size, $\Delta D_i$ (mm) is the $i$th bin width diameter interval. The sampling interval $\Delta t$ was fixed to 1-min to ensure there is a sufficient number of 140 measurements to compute a reliable DSD, which is also consistent with previous studies (Bringi et al., 2011; Ji et al., 2019). The terminal velocity of raindrops was computed by Gunn and Kinzer (1949):

$$V(D) = 9.65 - 10.3\exp(-0.6D_i) \tag{2}$$

The disdrometers measure the DSDs with a 1-min sampling interval. The Thies and Parsivel disdrometers measure the terminal velocity of raindrops and so they can classify precipitation particles based on the velocity-diameter relationship. Only 145 those measurements classified as liquid rain were used in this analysis. The DSDs were fitted to a normalised gamma drop size distribution using the procedure given in Bringi et al. (2003), where the normalised gamma DSD is given by:

$$N(D) = N_w f(\mu) \left(\frac{D}{D_m}\right)^\mu \exp\left[-(4+\mu)\frac{D}{D_m}\right] \tag{3}$$





where

$$f(\mu) = \frac{6}{4^4} \frac{(\mu+4)^{\mu+4}}{\Gamma(\mu+4)} \tag{4}$$

where $N_w$ (m$^3$ mm$^{-1}$) represents the normalised intercept parameter, $D_m$ (mm) is the mass-weighted mean diameter, and $\mu$ is the shape of the distribution. $D_m$ is related to $D_0$ (median volume diameter) for a gamma DSD with:

$$\frac{D_0}{D_m} = \frac{3.67 + \mu}{4 + \mu} \tag{5}$$

From the above analysis, the parameters $N_w$, $D_0$ (or $D_m$) and $\mu$ were retrieved for each 1-min measured DSD. Then, Eq. (3) was used to compute the theoretical DSD, which was used as input to a T-matrix scattering model developed by Mishchenko

(2000) and adapted to compute all the different polarimetric weather radar measurements, including $Z_H$ and $Z_{DR}$, which are both used in this analysis. The scattering simulations were performed using the following assumptions: (i) the raindrop shape model from Thurai et al. (2007) (their Eq. (2) for $D > 1.5$ mm, their Eq. (3) for $0.7 <= D <= 1.5$ mm, spherical raindrops otherwise); (ii) no canting angle distribution; (iii) maximum diameter for the integration fixed to $3D_0$; (iii) temperature of 10 °C, radar wavelength of 5.3 cm and elevation angle of 0 degrees.

## 3  Methods

### 3.1  Offset detection and monitoring of $Z_{DR}$ using vertical profiles

The overall system bias (or offset) in $Z_{DR}$ can be estimated using VPs taken in light rain as proposed by Gorgucci et al. (1999). The VPs represent averaged observations of the 360 vertical rays, reducing the variance in $Z_{DR}$ caused by the symmetry axis and the variety of shapes of the raindrops. Then, the premise of this method is to use VPs related to light rain, where a deviation

from 0 dB in the rain region of the VPs can be set as the $Z_{DR}$ offset.

The offset on $Z_{DR}$ can be detected and corrected by an automated operational procedure as follows:

1.  It is necessary to detect the rain region on the VPs; this can be done through a ML detection algorithm (e.g. see Sanchez-Rivas and Rico-Ramirez (2021)) and then setting the ML bottom as a boundary. Values on the VPs below this height are likely related to precipitation in the liquid phase.

2.  Once the rain region is identified on the VPs, thresholds related to light rain are set, and only VPs containing two or more consecutive bins of $Z_{DR}$ having corresponding values of 5 dBZ $< Z_H < 30$ dBZ and $\rho_{HV} > 0.98$ are kept for further calculations.

3.  The $Z_{DR}$ offset is calculated for each VP related to light rain using the following expression:

$$Z_{DR}^{O_{VP}} = \frac{1}{n} \sum_{i=1}^{n} Z_{DR_i} \tag{6}$$





where $i$ represents a bin along the VP, $n$ the number of bins below the melting layer and avoiding clutter echoes, $Z_{DR}^{O_{VP}}$ is the offset calculated from the vertical profile and $Z_{DR_i}$ represents the bins of $Z_{DR}$ below the ML. Note that $n$ includes bins from different azimuths. If $Z_{DR}^{O_{VP}}$ is different from 0 dB, then $Z_{DR}$ needs to be calibrated.

4. Finally, $Z_{DR}$ PPI measurements at different elevation angles can be corrected using:

$$Z_{DR}^{Oc} = Z_{DR}^{m} - Z_{DR}^{O_{VP}} \tag{7}$$

where $Z_{DR}^{Oc}$ is the offset-corrected differential reflectivity, $Z_{DR}^{m}$ is the differential reflectivity measured by the radar and $Z_{DR}^{O_{VP}}$ is the offset calculated from the vertical profiles.

In Figure 2, the left panel shows VPs of $Z_{DR}$ in a height-versus-time-plot related to a rain event recorded by the Chenies radar, whereas the right panel shows a single VP version taken from the same event. Figure 2 shows a persistent offset in $Z_{DR}$ as the values of this variable in the rain medium are deviated from 0 dB below the bottom of the bright band ($BB_{bottom}$). The

figure also shows that the first kilometre of the VPs is contaminated with spurious echoes, hence all bins below this height are discarded from the analysis. As shown in the right panel of Figure 2, the VP of $Z_{DR}$ (representing the mean value of 360 rays) and its standard deviation (blue area) is steady in the rain region (below 1.5 km), but it turns noisy above the ML bottom (grey dotted line), showing a higher standard deviation. For this particular VP, the computed offset in $Z_{DR}$ is -0.37 dB; this value is consistent throughout the whole event.

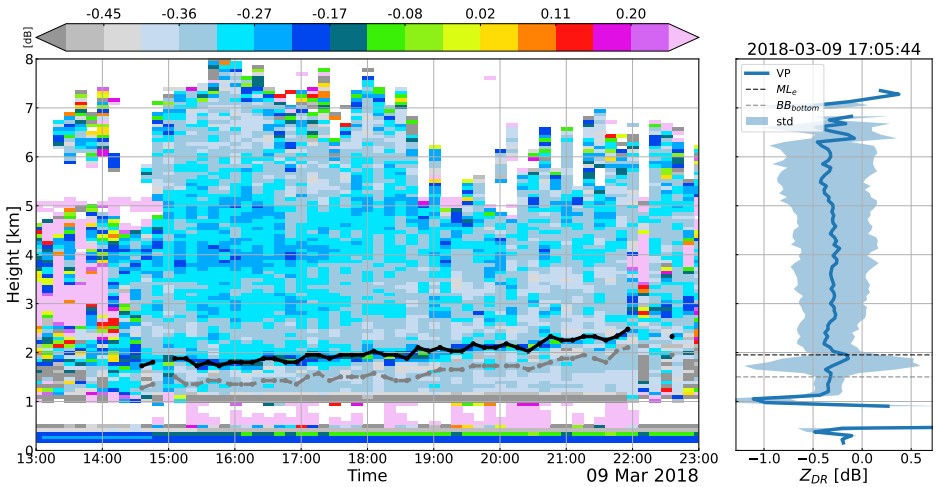

**Figure 2.** Collection of $Z_{DR}$ VPs for a rain event recorded by the UKMO Chenies radar. The left panel displays VPs in a height-versus-time plot along with the melting level (MLe) and the bottom of the melting layer ($BB_{bottom}$). The right panel depicts a single VP and its standard deviation (std).



## 3.2 Offset detection and monitoring of $Z_{DR}$ using a quasi-vertical profiles approach

The QVPs of polarimetric variables provide insight into the evolution and structure of rain events through time, thus enabling monitoring the calibration of the radar variables. Hence, we propose a method to estimate the $Z_{DR}$ offset that can be applied to QVPs generated from scans with elevation angles of around 10° and collected during light rain events. The rationale supporting the proposed method is described as follows:

a. The rain region within the QVPs of $Z_{DR}$ is mostly uniform when the profiles are generated from data collected in light rain and near the radar as this region represents averaged observations of oblate-spheroid raindrops. Figure 3 portrays the radar coverage of two PPI scans at different elevation angles recorded by one of the radars. It can be seen that birdbath scans (and subsequent VPs) capture uniformly the rain region (between 1 km and 2.5 km in height) developing above the radar (Figure 3(a)). Similarly, the rain region below the bright band (located at 2.5 km in height) is mostly homogeneous when using a PPI from an angle elevation of 9° (Figure 3(b)).

b. The intrinsic value of $Z_{DR}$ for angles below 90° and collected in light rain is different from zero. Also, it is elevation-dependent, as demonstrated by Bringi and Chandrasekar (2001) and formulated by Ryzhkov et al. (2005a) as:

$$Z_{dr}(\theta) \approx \frac{Z_{dr}(0)}{\left[Z_{dr}^{1/2}(0)\sin^2\theta + \cos^2\theta\right]^2} \tag{8}$$

where $Z_{dr}(0)$ and $Z_{dr}(\theta)$ represent the differential reflectivity in linear scale at elevation angles of 0° and $\theta$°, respectively. Figure 4 displays the theoretical variation of $Z_{DR}$ with elevation angle. It can be seen that the variation of $Z_{DR}$ as a function of the the elevation can be neglected for elevations below 10°, in fact using Eq. (8) for $\theta = 10°$ results in $Z_{dr}$ values very close to one:

$$Z_{dr}(\theta = 10°) \approx 0.968 Z_{dr}(\theta = 0°) \text{ [dB]} \tag{9}$$

Hence, $Z_{DR}$ radar measurements collected at elevation angles below 10° are similar to those collected at lower elevation angles and so they do not add additional uncertainty to the offset correction method.

Based on the premises described above, we propose an operational method to detect and correct the $Z_{DR}$ offset using QVPs, and it is described next.

1. As in the VPs method, the rain region is identified in the QVPs using a ML detection algorithm to set the ML bottom as a boundary. Values below this height are likely related to precipitation in the liquid phase. Additionally, a maximum height limit of 3 km is set to this ML bottom boundary to reduce the effect of the varying diameter inherent to the generation process of the QVPs.

2. Using the data sets described in section 2.2, we computed the mean dependencies of $Z_H - Z_{DR}$ but limited to a range related to light rain (0-20 dBZ), as shown in Figure 5. Then, we compute the average value of these curves to include data from different disdrometer models and different locations into the analysis, thus setting 0.18 dB as the mean intrinsic value of $Z_{DR}$ in light rain at ground level.





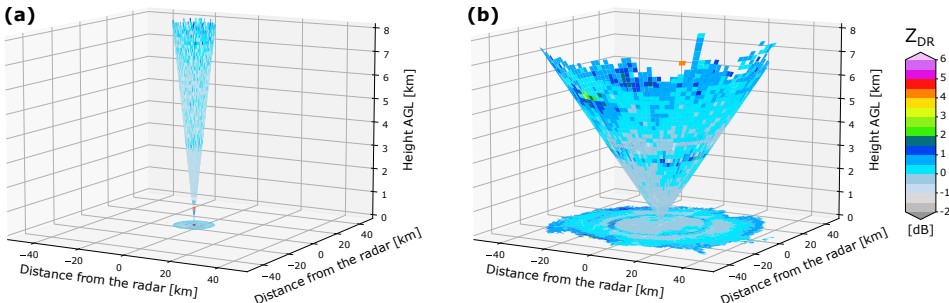

**Figure 3.** Representation of the radar conical coverage using (a) a birdbath scan, useful to build VPs and (b) a $9°$ PPI scan, used to generate QVPs.

3. Thresholds are set to detect QVPs related to light rain and discard bins within the QVPs related to mixed-phase precipitation. Thus, only QVPs containing four or more consecutive bins of $Z_{DR}$ with corresponding values of $0\ \text{dBZ} < Z_H < 20\ \text{dBZ}$ and $\rho_{HV} > 0.985$ on the QVPs of $Z_H$ and $\rho_{HV}$ respectively, are kept for further calculations. Note that the threshold set for $Z_H$ is the same as the range selected in the disdrometer data, whereas the threshold set for $\rho_{HV}$ is more strict than in the method based on VPs, this discard bins within the QVPs not related to light rain.

4. The average value of $Z_{DR}$ is computed, calculating one value per QVP related to light rain:

$$Z_{DR}^{O_{QVP}} = \left(\frac{1}{n}\sum_{i=1}^{n} Z_{DR_i}\right) - 0.18\ \text{dB} \tag{10}$$

5. Finally, $Z_{DR}$ measurements can be corrected by:

$$Z_{DR}^{Oc} = Z_{DR}^{m} - Z_{DR}^{O_{QVP}} \tag{11}$$

where $Z_{DR}^{Oc}$ is the offset-corrected differential reflectivity, $Z_{DR}^{m}$ is the differential reflectivity measured by the radar and $Z_{DR}^{O_{QVP}}$ is the offset calculated from the quasi-vertical profiles.

Figure 6 shows QVPs of $Z_{DR}$ generated from data collected by a C-band weather radar. It can be seen that there are clear signatures of the melting layer on the QVPs that are useful to classify the hydrometeors phase. In the right panel of Figure 6, a QVP of $Z_{DR}$ taken from the same event shows that the standard deviation of the averaged values used to generate the profile is smaller on the rain region compared to the standard deviation within and above the ML (above 1.3 km). For this example, the averaged value of $Z_{DR}$ in the rain region is -0.21 dB that along with the computed intrinsic value of $Z_{DR}$ (-0.18 dB), results in an offset of -0.39 dB, which is very close to the offset calculated using the VPs method (-0.37 dB).

## 4 Long-term monitoring of the $Z_{DR}$ calibration

The $Z_{DR}$ offset is prone to fluctuate during short periods of time (daily or even hourly). These fluctuations can affect the rainfall estimations based on radar measurements; hence the $Z_{DR}$ offset needs to be monitored, and $Z_{DR}$ measurements need





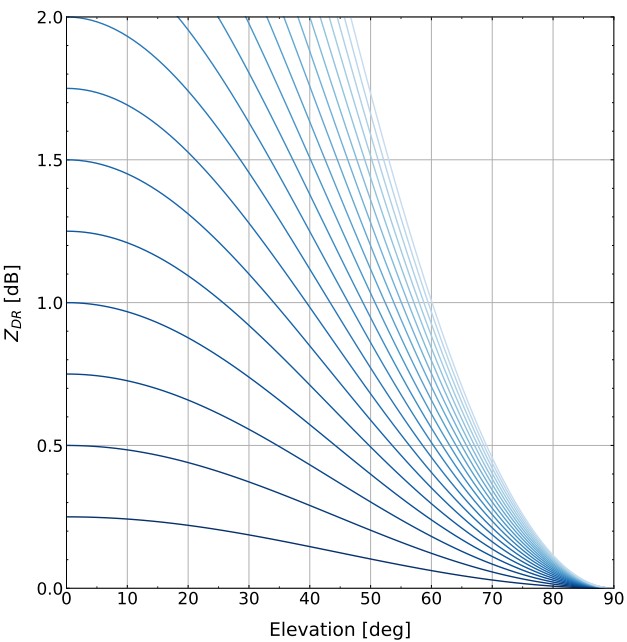

**Figure 4.** Theoretical dependencies of $Z_{DR}$ at different elevation angles.

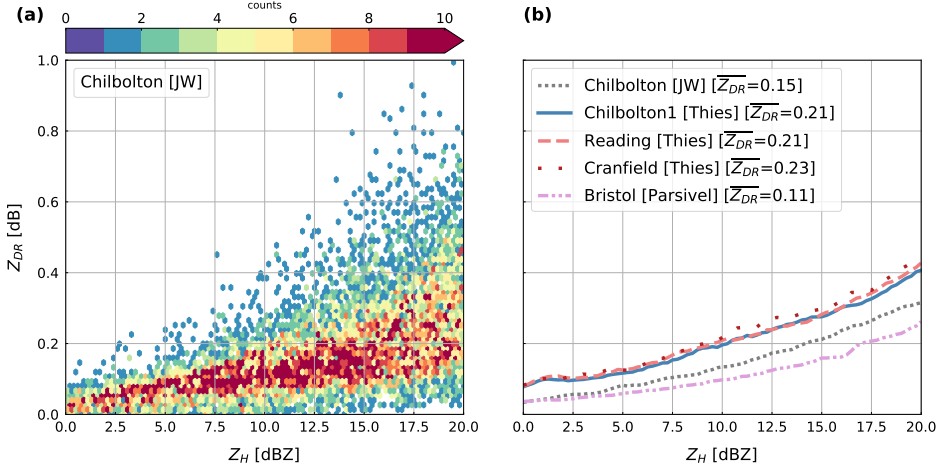

**Figure 5.** (a) $Z_H$-$Z_{DR}$ dependencies simulated using Chilbolton disdrometer data set; (b) Mean $Z_H$-$Z_{DR}$ dependencies simulated at several disdrometers locations.





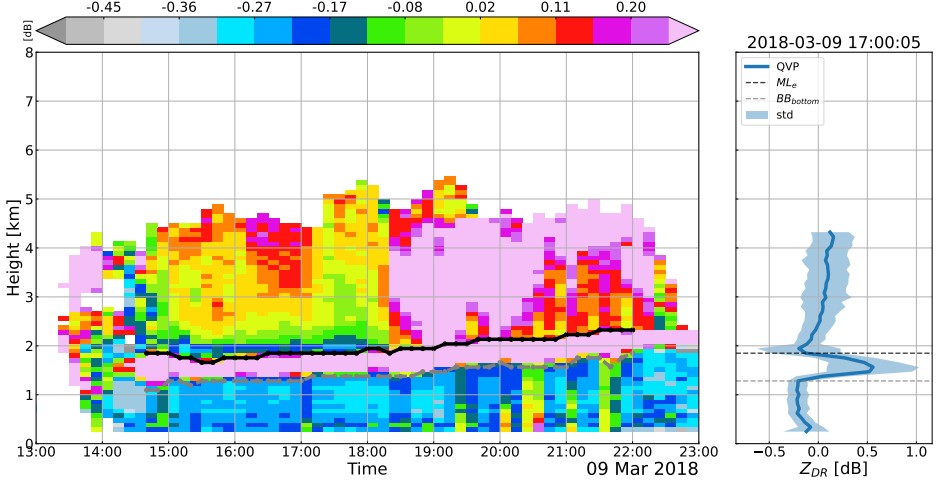

**Figure 6.** Collection of $Z_{DR}$ QVPs for a rain event recorded by the UKMO Chenies radar. The left panel displays QVPs in a height-versus-time plot along with the melting level (MLe) and the bottom of the melting layer ($BB_{bottom}$). The right panel depicts a single QVP and its standard deviation.

to be calibrated accordingly. We applied both $Z_{DR}$ offset-correction methods on two radar data sets throughout one year of precipitation events to compare the results of the $Z_{DR}$ calibration. The QVPs-based approach will be assessed by comparing its results with the 'true offset' computed from the VPs-based method since this method is widely accepted and proven effective, as described in the literature review. Additionally, both methods will be compared to independent measurements provided by

the disdrometers.

## 4.1 Validation of the QVPs-based approach using birdbath scans

We processed the radar data sets to generate VPs and QVPs of polarimetric variables for one year of radar data, as described in Section 2.1. Then, both the VPs-based and QVPs-based methods were applied to calibrate and monitor $Z_{DR}$.

Figure 7 shows the temporal variation of the $Z_{DR}$ offset for the two radars used in this work. For the Chenies radar data
set (Figure 7(a)), it can be seen that the offset in $Z_{DR}$ computed using the birdbath method fluctuates between -0.2 dB and -0.7 dB during most of the year. During February 2018, filters were installed at the Chenies radar, introducing a variation on the radar calibration that can be observed at this period (Timothy Darlington, Met Office, personal communication, 2021). The proposed method based on QVPs proves to be effective as the $Z_{DR}$ offset values are similar to those calculated using VPs. For the Dean Hill radar data sets, the $Z_{DR}$ offset varies on a wider range, but as above, the computed offset is similar on
both methods. Similarly, an upgrade implemented on the Dean Hill radar during October 2018 modified the $Z_{DR}$ calibration (Timothy Darlington, Met Office, personal communication, 2021), changing from -0.2 dB to 0.5 dB around this time of the year. These results confirm the efficacy of the proposed method based on QVPs, achieving an accuracy of $\pm0.1$ dB compared to the





method based on VPs. Additionally, the figure shows a particular rain event in a zoomed up window for a deeper visualisation of the calibration methods. This figure shows that both methods produce similar results.

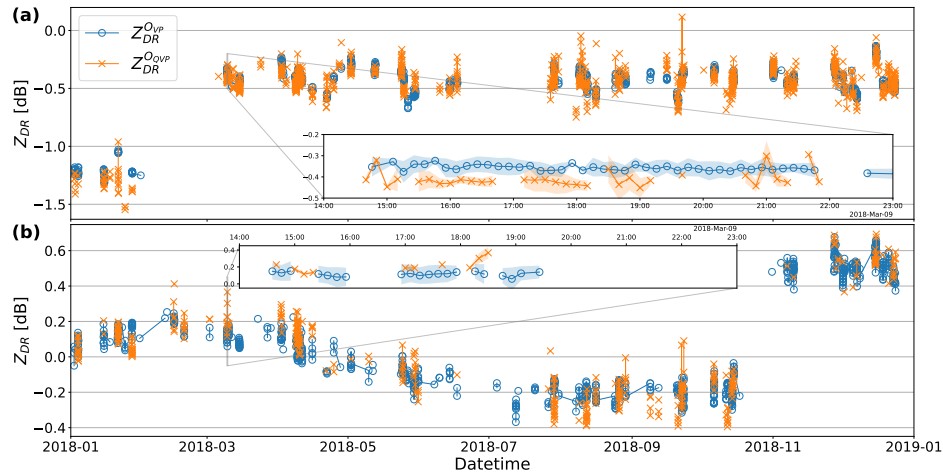

**Figure 7.** Temporal variation of the $Z_{DR}$ offset on two weather radars during one year of rain events. The top panel shows the variation on the Chenies radar site, whilst the bottom panel depicts the offset variation at the Dean Hill radar site. The case of a rain event on 09 March 2018 is zoomed in on both panels for an in-depth examination.

Finally, we evaluate the outputs of each method for the two different radar sites using different metrics like the correlation coefficient (r), the mean absolute error (MAE), and the root mean squared error (RMSE). To effectively assess the performance of the QVPs-based approach and its temporal variation, each offset value is extended until a new offset is computed, e.g., in Figure 7(a), for the case on 09-March-2018, the VPs-based method yields a constant offset value of around -0.36 dB between 19:05 and 20:45, whereas for the QVPs-based method there are only three data points for the same period of time, but the offset

is similar at those points in time (-0.39 dB, -0.37 dB and -0.36 dB). Figure 8 shows the correlation between the outputs of both methods, where a global accuracy of the proposed method ($\pm 0.1$ dB) is in good agreement with the 'true offset' computed from birdbath scans.

## 4.2  Differential reflectivity comparison using radar and disdrometers

Several validation procedures of the proposed method for correcting the $Z_{DR}$ offset were performed utilising the disdrometer

data sets described in Section 2.2. The fitted normalised gamma DSDs allow the estimation of the reflectivity and the differential reflectivity at ground level, enabling the validation of the QVPs-based method.

First, we compare the radar calibrated $Z_{DR}$ measurements by the two approaches described in the previous sections at the disdrometer locations. Only individual radar bins exactly over the corresponding disdrometers locations are considered for comparison. Based on the distance between the radars and the disdrometers, we link the Cranfield and Reading disdrometers

to the Chenies radar, whereas the Chilbolton1 disdrometer will be compared to the Dean Hill radar. Note that the disdrometer





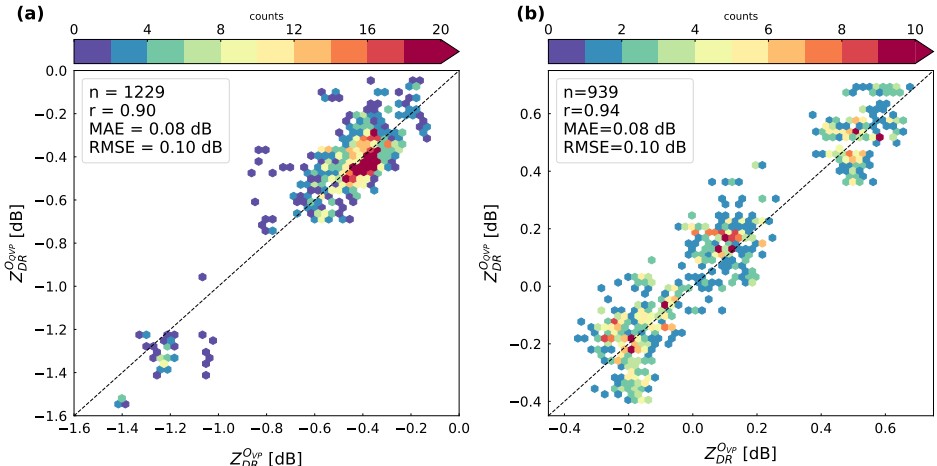

**Figure 8.** $Z_{DR}$ offset comparison on two radar data sets of the QVPs-based method versus the method based on birdbath scans. The scatter density plot shown in (a) provides metrics for evaluating the methods applied to the Chenies radar data set; whereas (b) shows the same as in (a) but for the Dean Hill radar data set.

located at Bristol was not used in this analysis because it is too far from both radar sites. In addition, we use the classifiers produced by the Thies disdrometers to evaluate radar data only related to liquid precipitation, as these disdrometers provide information about the rain type and intensity.

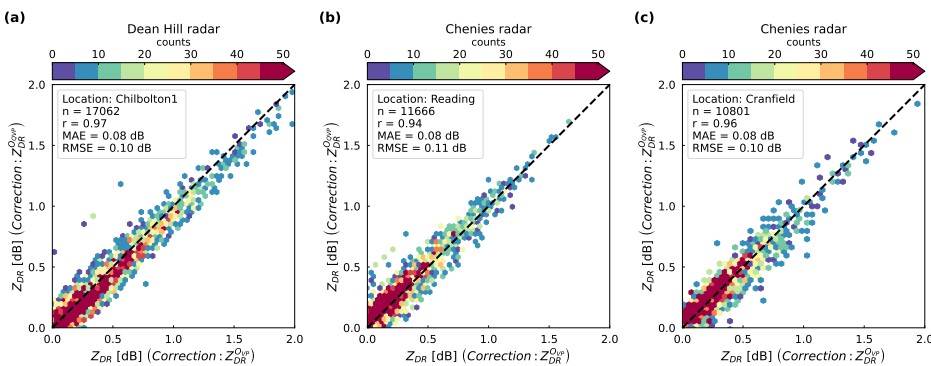

**Figure 9.** Correlation between two different approaches to correct the $Z_{DR}$ offset. Each scatter density plot represents $Z_{DR}$ measured by the radar at different locations and filtered using precipitation and intensity classifiers gathered from disdrometers.

Figure 9 shows the scatterplots using both methods used to correct the $Z_{DR}$ offset at the disdrometer locations. For the
Chenies radar data, we applied Eq. (7) and (11) to correct the offset in PPI scans taken at $0.5°$ elevation angle, whilst for the Dean Hill data, we applied the same equations to correct the offset but on PPI scans taken at $2°$ elevation angle as lower





elevations are beam-blocked or clutter-contaminated. The proposed approach based on QVPs proves effective as an accuracy $\sim 0.1$ dB is achieved in all analysed cases.

In addition, we compare the polarimetric variables measured by the radar with the variables derived from disdrometer DSDs.

We discard data not related to liquid precipitation by using the classifiers available on the disdrometer data sets and using only radar data with corresponding values of $\rho_{HV} \geq 0.98$. As described in Section 2.1, algorithms for removing non-meteorological echoes and for correcting the signal attenuation are applied to radar data sets when appropriate. Regarding the disdrometer data sets, we applied a moving average filter (window size = 5) to reduce data fluctuations due to the higher time resolution of the disdrometer data than radar data sets. Furthermore, to include data collected by the CFARR Chilbolton disdrometer (model

Joss–Waldvogel, not capable of classifying the rain type), we used the classification from the Thies disdrometer (Chilbolton1) to discard data from the former not related to rain, as these two disdrometers are close to each other (just a few meters apart). Figure 10 shows the comparison between $Z_{DR}$ measured by both radars and $Z_{DR}$ derived from disdrometer observations located near the radar sites and collected throughout one year of precipitation events. The top row of Figure 10 shows calibrated radar $Z_{DR}$ measurements using the method based on VPs and $Z_{DR}$ derived from disdrometers. Similarly, Figure 10(b) shows

disdrometer and radar $Z_{DR}$ measurements. The $Z_{DR}$ radar observations are offset-corrected using the proposed approach based on QVPs. The scatter plots are very similar to those obtained using the $Z_{DR}$ correction method based on VPs, thus reaffirming the good performance of the proposed method. As mentioned above, scans taken at different elevation angles are used on each radar to capture the precipitation occurring above the disdrometer, adding some uncertainty to the interpretation of these results. An in-depth analysis of these results is provided in Section 5.

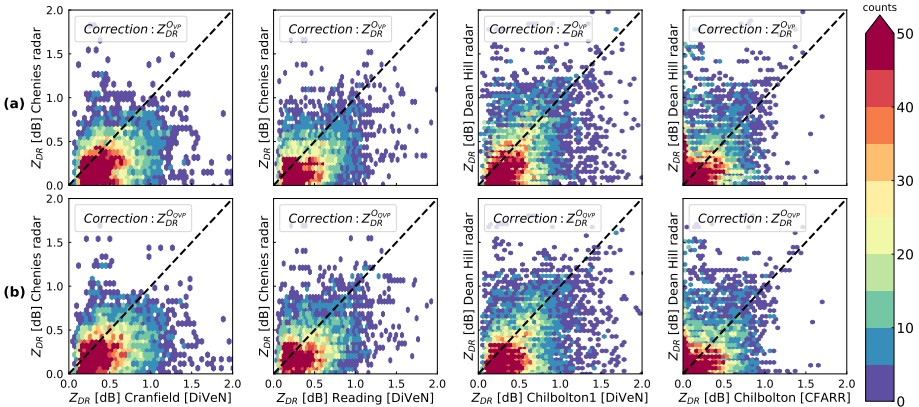

**Figure 10.** Scatterplots between radar and disdrometer $Z_{DR}$ measurements at several locations: (a) shows scatter density plots of $Z_{DR}$ offset-corrected using VPs at two different radar sites versus $Z_{DR}$ derived from disdrometer data; (b) shows the same as in (a) but the radar $Z_{DR}$ measurements are calibrated applying the QVPs-based method.



### 4.2.1  Case study: 24 May 2018

Figure 11 portrays a rain event recorded by the Dean Hill radar at an angle elevation of $2°$ and data from the disdrometers described above. The top panel shows a good agreement between the radar reflectivity and the reflectivity derived from disdrometer DSDs as there is a similar trend on all data sets. Overall, the correlation of $Z_H$ for the whole year of data between both radar data sets and several disdrometers is $\geq 0.75$ in all the cases (graph not shown). On the other hand, the bottom panel of Figure 11 illustrates the radar differential reflectivity offset-corrected by both the birdbath-scans-based method and using the proposed QVPs-based approach; differential reflectivity derived from DSD data from two nearby disdrometers is also displayed. It can be seen that the proposed method is in good agreement with the 'true offset' computed from scans collected at vertical incidence as a maximum difference of $\pm 0.1$ dB is observed. Even more, both methods are consistent with the data derived from the two disdrometers. Further discussion of these results is provided in Section 5.

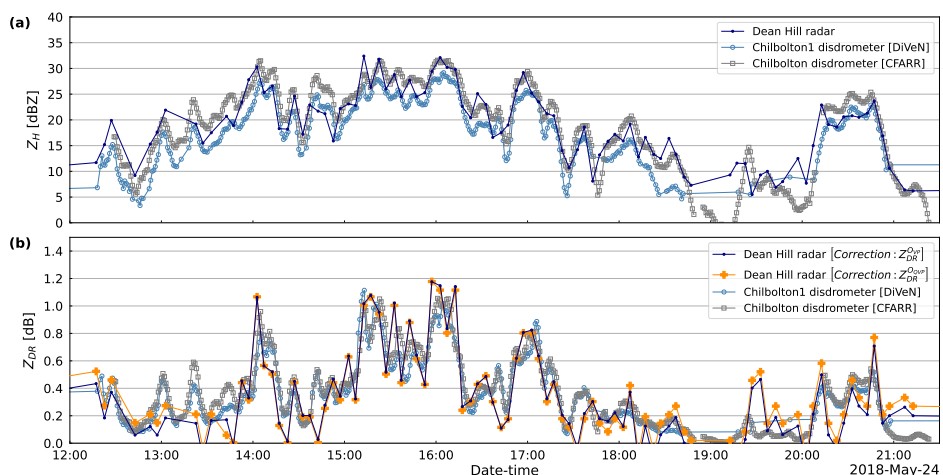

**Figure 11.** Time-series of disdrometer and radar data related to a precipitation event registered at south England; (a) Reflectivity ($Z_H$) simulated from disdrometer DSD data at two nearby locations and $Z_H$ measured by a C-band weather radar at an angle elevation of $2°$; (b) Differential reflectivity ($Z_{DR}$) measured by the radar and offset-corrected using two different approaches and $Z_{DR}$ simulated from two disdrometers.

## 5  Discussion

We investigate the use of quasi-vertical profiles of polarimetric variables to estimate and monitor the overall system bias (or offset) in the differential reflectivity $Z_{DR}$. Although several sources of error affect this variable, we focused on detecting and correcting the overall system bias. It is important to keep $Z_{DR}$ calibrated as this variable is a crucial input to hydrometeor classification methods (Al-Sakka et al., 2013; Park et al., 2009), attenuation corrections schemes (Bringi et al., 2011; Gou et al., 2019) or QPE algorithms (Chandrasekar and Bringi, 1988; Cifelli et al., 2011; Giangrande and Ryzhkov, 2008; Ryzhkov





et al., 2005b; Vulpiani et al., 2009). Previous research has demonstrated that keeping $Z_{DR}$ well-calibrated, i.e., the bias is less than 0.2 dB, generates accurate and reliable radar products (Ryzhkov et al., 2005a).

First, we implemented an operational procedure to detect the offset based on the rationale proposed by Gorgucci et al. (1999) using measurements taken at vertical incidence. The key assumption of this method is that the intrinsic value of $Z_{DR}$ in light

rain is 0 dB; values deviated from this criterion can be related to the $Z_{DR}$ offset. A similar rationale has been implemented on several radar campaigns confirming its reliability by keeping the $Z_{DR}$ offset below 0.1 dB (Bechini et al., 2002; Frech and Hubbert, 2020; Gourley et al., 2009; Louf et al., 2019). We use vertical profiles of polarimetric variables to implement our approach and proposed thresholds to discard data unrelated to light rain. An important condition before applying this approach relies on detecting the melting layer, as this enables the identification of echoes related to liquid precipitation.

Figures 3 and 4(a) illustrate important aspects of this method. It is important to remark that the VPs represent averaged measurements taken while the radar rotates through $360°$, as depicted in Figure 3(a). Then, a collection of VPs related to a precipitation event recorded by a C-band weather radar is shown in the left panel of Figure 2. This plot shows that a major limitation of this approach is when the melting layers are at a relatively low height, and bins related to light rain are not available. Furthermore, in our data sets, the first kilometre of the VPs is contaminated by spurious echoes, thus reducing the

number of available bins to implement this method, e.g., at times between 15:00 and 17:00 hours, the melting level bottom (upper boundary of the rain medium) ranges between 1.2-1.3 km; hence only a few bins are available within the VPs to compute the offset. It also notable that $Z_{DR}$ values within and above the melting layer are quite unrelated to the $Z_{DR}$ offset computed from values taken in the rain medium. As shown in the right panel of Figure 2, the standard deviation (blue area) increases within and above the ML due to the presence of mixed-phase particles, snow or ice; hence complicating the estimation of the

$Z_{DR}$ offset using such meteorological targets.

These conditions restrict the implementation of the method based on VPs and, coupled with mechanical restrictions of some weather radars that hamper performing birdbath scans, had lead to developing alternative methods to correct the offset in $Z_{DR}$ (e.g., see Bechini et al. (2008); Bringi et al. (2006); Chu et al. (2019); Ferrone and Berne (2021); Giangrande and Ryzhkov (2005); Gourley et al. (2006); Holleman et al. (2010); Richardson et al. (2017); Ryzhkov et al. (2005a), among others). Thus,

we present a novel approach to detect and correct the offset in the radar $Z_{DR}$ that can be applied to quasi-vertical profiles of polarimetric variables built from PPI scans taken at elevation angles of around $10°$.

As described in Section 3.2, this method is similar to the method based on VPs, but the intrinsic value of $Z_{DR}$ in light rain is derived using disdrometer data instead. We limit our method to elevation angles of around $10°$ because using much lower angles to generate QVPs yields vague polarimetric signatures. On the contrary, using greater elevation angles adds uncertainty

to the premise described in Eq. (9) and illustrated in Figure 4, where $Z_{DR}$ measurements much vary when the elevation angles start to increase. It is also important to discuss the imposed limit of 3 kilometres to apply our method: as shown in Figure 3(b), the coverage of the PPI scans at $9°$ elevation angle captures a mostly uniform volume in the rain region (below 2.5 km). For this elevation angle and a height of 3 km, the base diameter of the cone is around 37 km. Hence we consider that the azimuthal averaging procedure to generate the QVPs below this height reduce deviations in $Z_{DR}$ and enables proper monitoring of the

calibration of this variable. Additionally, we define thresholds to discard values within QVPs not related to light rain, e.g.,





Figure 6 shows a collection of QVPs related to a rain event. Most of the QVPs show a constant value of $Z_{DR}$ below the ML, whilst outlier values can be discarded by checking their corresponding values on the QVPs of $Z_H$ and $\rho_{HV}$ (plots not showed). This plot also shows that the values of $Z_{DR}$ outside the rain region are loosely correlated to the $Z_{DR}$ offset; hence we consider that using meteorological targets like snow or ice is not feasible for our method. Finally, we used data derived from three

different models of disdrometer and located at different locations to produce robust $Z_H - Z_{DR}$ dependencies that are used to estimate the intrinsic value of $Z_{DR}$ in light rain, as shown in Figure 5. This value is applied in Eq. (10), later used to estimate the offset in the QVPs of $Z_{DR}$.

As mentioned above, there are several methods proposed in the literature to detect and monitor the offset in $Z_{DR}$ not based on VPs and making use of other atmospheric targets. But most of them are limited by radar mechanical restriction to point

vertically; hence comparing its results with the 'true offset' computed from VPs is not feasible. Even more, none of them uses QVPs of polarimetric variables (a more broadly used tool in recent years) to estimate the $Z_{DR}$ offset; hence, we carry out a thorough comparison between our proposed method and the method based on birdbath scans.

Figures 7 and 8 show the good agreement between both methods: the proposed method based on QVPs achieves an overall accuracy of around 0.1 dB on the calibration of $Z_{DR}$ compared to the method based on VPs. A few data points exhibit larger

variation, but this is mainly caused by unclear polarimetric signatures of the ML, misleading the ML detection algorithm and, thus, the classification of the particles in the liquid phase. The good performance of the method based on QVPs is confirmed in Figure 9, where we evaluate only data classified by the disdrometers as related to light to moderate rain rates. These results are in good agreement with the required accuracy of 0.2 dB established by Ryzhkov et al. (2005a) to generate reliable quantitative precipitation estimates using polarimetric weather radar data. The proposed method proves effective on the calibration of $Z_{DR}$

compared to other methods not based on VPs like Ryzhkov et al. (2005a) ($\approx 0.2$ dB), Giangrande and Ryzhkov (2005) ($\approx 0.3$ dB), or Bechini et al. (2008) ($\approx 0.1$ dB), improving the accuracy on the $Z_{DR}$ calibration.

Finally, Figures 10 and 11 portrays a comparison between radar and disdrometer data. Before analysing these results, it is important to keep in mind factors such as (i) the spatial distribution of radar measurements and the well-known discrepancy when comparing it to a fixed point location, (ii) the impact of the signal attenuation in $Z_H$ and $Z_{DR}$, (iii) the distance between

the radars and the disdrometers, (iv) the use of PPI scans collected at higher elevation containing issues related to beam blockage or clutter contamination and (v) the different temporal resolution of each device. It is important to remark that the comparison shown on both plots is made using disdrometer data smoothed using a moving average filter (window size = 5 [min]). Regarding $Z_{DR}$, the global correlation between the data sets is acceptable, considering the factors mentioned above. Furthermore, we compared the disdrometer $Z_{DR}$ with the radar $Z_{DR}$ but without applying the offset correction procedure,

and we observed bigger discrepancies between data sets, reaching differences in the order of 1 dB (plots not shown). A higher correlation is observed when comparing data from the Dean Hill radar and the Chilbolton1 disdrometer. This can be explained by the distance between devices ($\sim 20$ km) and the hydrometeor classification available in this disdrometer. The case study shown in Figure 11 confirms the good performance of the proposed method to correct the $Z_{DR}$ offset.





## 6 Summary and Outlook

In this work, we have evaluated different methods for monitoring the calibration of the radar differential reflectivity ($Z_{DR}$). The most important findings are summarized herein:

- We implemented a well-known methodology to calibrate and monitor the $Z_{DR}$ offset using vertical profiles of polarimetric variables. The method is based on the intrinsic value of $Z_{DR}$ collected in light rain.

- We proposed a novel, operational method to calibrate $Z_{DR}$ using quasi-vertical profiles of polarimetric variables. There
are two main advantages of this approach over the method based on VPs: (i) it can be implemented on radars not capable of performing scans at vertical incidence, (ii) it can detect the offset on rain events not happening exactly above the radar.

- Although the intrinsic value of $Z_{DR}$ required to implement the method based on QVPs was calculated using disdrometers at different locations and different types and models of disdrometers; this value could be re-calculated using disdrometer data near to the radars that want to be calibrated.

- We applied both methods over one year of precipitation events collected by two C-band weather radars. The proposed method to detect the offset in $Z_{DR}$ using QVPs was compared against the 'true offset' computed from the VPs. We observed an excellent agreement between both methods, as the MAE and the RMSE are within $\pm0.1$ dB.

- We observed that the $Z_{DR}$ offset varies daily, or even hourly, hence it is necessary to monitor the calibration of $Z_{DR}$ to generate reliable polarimetric radar products.

- We compared $Z_{DR}$ measured by the radars with $Z_{DR}$ derived from disdrometer measurements, obtaining a good agreement between the various data sets. However, it is important to keep in mind that we also applied other procedures like the signal attenuation correction to the radar observations. Even more, the distance between some disdrometers and the radars increases up to 40 km in some cases. These factors add some uncertainty to the interpretation of the results.

- The proposed method using QVPs generated from PPIs proved to be effective for calibrating and monitoring the radar
differential reflectivity. Our results are similar to those produced by the traditional method that uses birdbath scans.

*Data availability.* Chenies C-band rain radar dual polarisation products are available on request from https://catalogue.ceda.ac.uk/uuid/bb3c55e36b4a4dc8866f0a06be3d475b; Dean Hill C-band rain radar dual polarisation products are available on request from https://catalogue.ceda.ac.uk/uuid/5b22789f362c43f3b3d1c65bc30c30ee; disdrometer data collected by the Chilbolton Facility for Atmospheric and Radio Research (CFARR) are available on request from https://catalogue.ceda.ac.uk/uuid/aac5f8246987ea43a68e3396b530d23e; DiVeN particle di-
ameter and fall velocity measurements are available on request from http://catalogue.ceda.ac.uk/uuid/001b9640fdb1453aa95a222ba423580e; disdrometer data collected at the UoB are available from the authors upon request.





*Author contributions.* DSR was responsible for carrying out the experiments, data analysis and writing of the paper. MARR provided supervision of the work and contributed to the writing of the paper.

*Competing interests.* The authors declare that they have no conflict of interest.

*Acknowledgements.* This work was carried out using the computational facilities of the Advanced Computing Research Centre, University of Bristol (http://www.bris.ac.uk/acrc/, last access: 05 June 2021)

*Financial support.* This research has been supported by the Mexican National Council for Science and Technology (CONACyT; grant no. 637289) and the Engineering and Physical Sciences Research Council (EPSRC; grant no. EP/I012222/1).





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
