# Peer review of "Calibration of radar differential reflectivity using quasi-vertical profiles"

_Atmospheric Measurement Techniques, 2021_

## Referee Comment (RC1)

Review of: "Calibration of radar differential reflectivity using quasi-vertical profiles", by Daniel Sanchez-Rivas and Miguel A. Rico-Ramirez

**General Comments:**

This manuscript offers a method for monitoring the ZDR offset of a dual-polarization radar using quasi-vertical profiles (QVP). The method is applied on C-band weather radars in light rain media. The authors suggest accuracy to O[0.1 dB], e.g., potentially in-line with 'bird-bath' calibration (natural media). There are two apparent justifications for this publication: its improvement compared to previous natural media efforts, and its QVP application towards these ideas.

The manuscript is not recommended for publication. The study is functional with elements similar to the typical AMT scope, but the reviewer finds low value in the 'new' concept/application. The use of intrinsic liquid properties for ZDR monitoring is well known, origins in low angles and selective ZDR averages (i.e., cell peripheries). This manuscript adds a "QVP"-wrapper aimed now at liquid media, yet lacks the physical underpinning as to why such methods would improve performance over a boilerplate practice to 'average ZDR in light rain'. These "QVP" concepts are evaluated against a modest dataset, but reads to the reviewer as motivated by convenience and applying a poorly-matched "QVP" concept (hammer looking for a nail?) in a less-behaved condition (light rain) to be 'novel'. Yet, it seems a straightforward evaluation of an existing snow QVP application (as less original as that seems) may have been far less controversial. The authors perhaps unintentionally increased their degree of difficulty (at least, to this reviewer), by leaving the reviewer questioning whether simpler, quicker, or (existing dry snow) options for targeted averaging may be equally/more effective.

Moreover, a central claim for this effort seems to follow its 'relative' calibration performance (oversold), esp. for "light rain". It is unlikely any 'natural' method can genuinely guarantee accuracy better than 0.2-0.3 dB – this has been well-argued by previous authors, including several cited; Prior efforts were rightfully cautious in their claims. Yes, some allowance can also be extended to older studies that are occasionally captives to their moment (i.e., radar technology improves with time  $\rightarrow$  better ability to target lighter rain, etc.). Nevertheless, the intrinsic "light rain" variability is significant and comes in many forms (not limited to):

- Capabilities to provide 'ground truth' (e.g., disdrometers as a poor light rain reference);
- What gets defined as 'light rain' (regional / physical process variability),
- How one identifies these regions with existing radar (Z calibration, etc.), and
- Location, radar sensitivity/quality, other vertical profile factors (e.g., evaporation, sorting, process) that undermine accuracy claims when averaging over regions.

For this reviewer, the authors have not demonstrated they built a better mousetrap. The reviewer understands there is an inevitable overconfidence (aka, marketing) in most manuscripts. However, "relative", not absolute calibration concepts are typically quite conservative, and it should be obvious that selective performance may be better under ideal

conditions. The authors' disdrometer image (Figure 5) alludes to some inherent variability in (surface, 'instantaneous') ZDR properties in "light rain" (aka, dynamic range of intrinsic ZDR > 0.6 dB). These depictions are consistent with discussions by Bechini et al., Ryzhkov et al., for what those authors expect from "light rain", or why "light rain" (generic) is less suitable than "dry snow" (see also, specific comment). Select locations (UK) experience different bulk microphysical expectations (e.g., propensity for widespread rainfall, stratocumulus), thus performances may reflect strong local process / natural advantages (e.g., contrast with "light rain" at the peripheries of thunderstorms).

Overall, one takeaway message is that this reviewer does not feel the authors have justified the "QVP" application as a genuine improvement over a generic "average" ZDR monitoring practice, for rain, snow or otherwise. Rather, the reviewer claim may be that "QVPs" in light rain are arguably far worse, given this form of averaging enables mixtures of less suitable profile properties that produce apparently viable "light rain" profiles. Why use a "QVP" process at all? Fundamentally, this is a reduction of information; Many previous studies speak to physical 'profile' issues convolved with "QVPs" and similar averaging, with even the QVP originators shifting to "CVPs" or other targeted averages – For example, ZDR should naturally evolve below the melting layer in response to processes such as sorting, evaporation, break-up, and/or other regime-averaging nuances (within event, or tropical vs midlatitude differences). This all points to why previous studies may have remained cautious in their claims on relative 'light rain' use and uncertainty, but also where QVP-ideas are suboptimal (esp. in rain, below cloud, etc.). The reviewer is questioning the need in using a QVP in these contexts if the QVP cannot be justified as out-performing any number of simpler, targeted ZDR averages of 'light rain' (if one is already thresholding regions loosely on Z, RHV regardless, you've already opened that echo classification bag once one introduced decision-tree thresholding for 'drizzle', etc).

**Specific Comment:**

Why do the authors use "light rain" for the "QVP"? Many efforts point to why they avoid light rain (see, Ryzhkov et al, discussions). Unfortunately, the reviewer might have been more amenable to an AMT manuscript that was simply a long-term validation for an existing 'dry snow' QVP concept. That is because most "QVP" concepts and ZDR calibration at higher tilts focus on the properties of lower density, dry aggregate snow as a claimed better-case media. They often note that the spatiotemporal averaging/variability is still a concern, but perhaps less in-cloud and widespread stratiform selective events. Overall, those rationale (e.g., Ryzhkov et al. and subsequent) reflect a somewhat different take on the role of higher tilts and the expected ranges for ZDR media at higher tilts. The current authors use expressions such as:

b. The intrinsic value of  $Z_{DR}$  for angles below 90° and collected in light rain is different from zero. Also, it is elevationdependent, as demonstrated by Bringi and Chandrasekar (2001) and formulated by Ryzhkov et al. (2005a) as:

$$Z_{dr}( heta) pprox rac{Z_{dr}(0)}{\left[Z_{dr}^{1/2}(0)\sin^2 heta + \cos^2 heta
ight]^2}$$

(8)

The reason Ryzhkov et al. give for higher tilts and dry snow is seemingly opposite to the current authors' logic – Ryzhkov argues dry snow has lower natural ZDR variability, and when these media are viewed from higher tilts (e.g., the eventual multiplier on ZDR in equation (9) would be closer to 0 instead of 1), the dynamic range of potential ZDR variability is low. When the underlying media experiences a wider range of variability, aka, light rain ranges from 0.1 dB to 0.6+ dB at typical trusted Z ranges, etc., this implies added uncertainty for any 'average' reference frame. These issues are at their most problematic at grazing angles, and possibly not preferable at lower altitudes (given evaporation, other profile physical processing that evolves ZDR below cloud). Thus, it is not immediately preferable (for their concepts) to have:

$$Z_{dr}(\theta = 10^{\circ}) \approx 0.968 Z_{dr}(\theta = 0^{\circ}) \text{ [dB]}$$

(9)

e.g., a high coefficient close to 1 is 'bad' for "light rain" in these contexts, b/c the intrinsic ZDR for Z ~ 15-20 dBz remains in those ranges from 0.1 to 0.6 dBz (aka, author Figure 5); This drives the potential uncertainty against the 'reference' ZDR, and one may be correcting by >0.3 dB quite often (perhaps this was worse in Oklahoma, where lower, unregulated use of Z caries a wider range of ZDR). High tilt intrinsic property sampling (if available, aka, 'birdbath' at its limit) acts to limit that range of possible ZDR  $\rightarrow$  better chance to accurately pinpoint ZDR. Thus, the authors' statement,

"Hence, ZDR radar measurements collected at elevation angles below  $10^{\circ}$  are similar to those collected at lower elevation angles and so they do not add additional uncertainty to the offset correction method."

... feels opposite this 'dry snow' rationale. This seems to be a question of whether the authors genuinely believe they can target low-ish variability ZDR 'drizzle' better than low-variability dry snow? This may be regional to the UK, e.g., stratocumulus w/drizzle, but may not seem as reasonable if painted with a US NEXRAD radar lens, as a separate example.

Even with light rain, these issues are likely worse than presented; For example, this effort has not fully discussed that the disdrometer (Parsivels, etc.) references are poor in light rain R < 1-3 mm/hr. It is unlikely most units capture light rain properties perfectly, esp. with assumptions made for disdrometer processing (a different subset of literature on Parsivel, 2DVD and other light rain comparisons). Dry snow media, similarly, has its own issues with identification, wavelength dependency, complications to "QVP" profiles from non-uniform beam-filling (at C-band, there is potentially intrinsic negative ZDR above the ML owing to non-uniform beam filling!). There is not a quick fix, unfortunately.

---

## Referee Comment (RC2)

**Review for AMT-2021-194**

**"Calibration of radar differential reflectivity using quasi-vertical profiles"**

**Daniel Sanchez-Rivas and Miguel A Rico-Ramirez**

This study proposes an operational method to estimate a systematic bias of radar differential reflectivity ( $Z_{dr}$ ) using quasi-vertical profiles (QVP). The authors compared the results of the proposed QVP method with those derived from vertical profiles (VP) and disdrometer data for one year period of 2018. They concluded that the new approach is consistent with the traditional method and is operationally applicable.

I think that this study is very important for radar quantitative precipitation estimation (QPE) based on polarimetric variables. However, I see a limitation of this study for an operational application. After reading the manuscript carefully, I found that the QVP method requires a disdrometer-derived  $Z_{dr}$  bias for light rain (e.g., 0.18 dB). This is a challenge where there is no disdrometer near radar sites. Additionally, using the disdrometer data in the QVP procedure (e.g.,  $Z_{dr}$  correction) affects an independent evaluation based on  $Z_{dr}$  derived from the disdrometer data (e.g., Fig. 10). My detailed comments are provided below.

**Major comments:**

**1. Title is misleading**

Just looking at the title, I started reading the manuscript with high hope to see how the QVP method can estimate a  $Z_{dr}$  bias. However, it turns out that the method needs a reference  $Z_{dr}$  value simulated from disdrometer measurements. This is a limitation for the operational estimation for most radar sites, particularly in the United States. I think that the author should include "disdrometer data" in the title.

**2. Independent evaluation**

Part of evaluation in this study is not independent. The disdrometer data used in the QVP procedure were also used in the evaluation (e.g., Figs. 10 and 11).

**3. Zh- Zdr dependence**

There is no  $Z_{h}$ -  $Z_{dr}$  dependence demonstrated in the manuscript. I think that the authors took simple averages of  $Z_{dr}$  values conditioned on  $Z_h$  values (0-20 dBZ) at each different disdrometer location.

**4. Discussion section**

The discussion section seems to be the summary of this study. Most of the paragraphs are summaries of the results presented in the figures described in the previous sections. I would like

to see actual discussions e.g., regarding any challenges or limitations (or sensitive factors) that can affect the accuracy of QVP method. Additionally, there is no "outlook" in the last section.

**Minor comments:**

1. Line 4 Maybe "light rain" instead of "rain?"

2. Line 4 Please replace "expected" with "desirable."

3. Line 95

Could the author specify the elevation angle of birdbath scans? Based on "averaging azimuthally," the elevation angle is not 90 degrees.

4. Line 169 Why not a "solid phase?" I think that  $Z_{dr}$  for solid phase should be reliable (for VP) if the authors avoid the melting layer (e.g., mixed phase) as seen in Fig. 2 (right).

5. Figure 2 The lines indicating the ML and ML bottom are different between right and left panels.

6. Line 217 What are "the mean dependencies?"

7. Line 219 Is the value 0.18 dB supposed to dynamically change depending on different event cases in an operational situation? Otherwise, is this value static?

8. Line 236 Please remove the negative sign in "-0.18 dB."

9. Line 239 Why does  $Z_{dr}$  offset fluctuate hourly? Is it a mechanical issue?

10. Figure 4 Please insert a legend for lines with different colors.

11. Figure 7

While values with VP look consistent, what is the reason of variations with the QVP method in the insets?

12. Line 293 Please replace "The top row of Figure 10" with "Figure 10(a)." 13. Line 344 Please provide more details about "vague polarimetric signatures."

---

## Author Comment (AC1)

**Interactive comment on "Calibration of radar differential reflectivity using quasi-vertical profiles". Response to Anonymous Referee 1**

Daniel Sanchez-Rivas and Miguel Angel Rico-Ramirez

October 9, 2021

We thank the reviewer for the insightful review of the manuscript and for the interesting feedback that surely will improve our work. In the following, we provide our answers (in blue) to the reviewer comments (in black), highlighting the key points of each paragraph (bold black).

**General Comments:**

This manuscript offers a method for monitoring the ZDR offset of a dual-polarization radar using quasi-vertical profiles (QVP). The method is applied on C-band weather radars in light rain media. The authors suggest accuracy to O[0.1 dB], e.g., potentially in-line with 'bird-bath' calibration (natural media). There are two apparent justifications for this publication: **its improvement compared to previous natural media efforts**, and its QVP application towards these ideas.

1. We consider that our method targeting **natural media** (i.e., light rain) is applicable when birdbath scans are not available, but **we are not suggesting that our approach is better** or should replace the well-known ZDR calibration method based on 90° VPs. In fact, the performance of our method is relative to the performance of the VP-based ZDR calibration method. The VP-based calibration method should be used where possible, providing that VP scans are available and there is rainfall above the radar. However, if this is not the case, our QVP-based ZDR calibration method is a very good alternative. As with any other method, ours faces advantages and disadvantages as discussed in the following paragraphs.

The manuscript is not recommended for publication. The study is functional with elements similar to the typical AMT scope, but the reviewer finds **low value in the 'new' concept/application**. The use of intrinsic liquid properties for ZDR monitoring is well known, origins in low angles and selective ZDR averages (i.e., cell peripheries). This manuscript adds a "QVP"-wrapper aimed now at liquid media, yet lacks the physical underpinning as to why such methods would improve performance over a boilerplate practice to 'average ZDR in light rain'. These "QVP" concepts are evaluated against a modest dataset, but reads to the reviewer as motivated by convenience and applying a poorly-matched "QVP" concept (hammer looking for a nail?) in a less-behaved condition (light rain) to be 'novel'. Yet, it seems a straightforward evaluation of an existing snow QVP application (as less original as that seems) may have been far less controversial. The authors perhaps unintentionally increased their degree of difficulty (at least, to this reviewer), by leaving the reviewer questioning whether simpler, quicker, or (existing dry snow) options for targeted averaging may be equally/more effective.

2. We consider that **this new application of the QVPs is needed** because several weather radar networks worldwide are leaning towards providing the QVPs as an operational product. We understand that the use of QVPs can be controversial due to the averaging process required for their construction. However, several works (see Allabakash et al. (2019); Griffin et al. (2018); Lukach et al. (2021); Ryzhkov et al. (2016); Trömel et al. (2019) for example) have demonstrated the ability of QVPs to detect the melting layer (ML), correct the vertical profile of reflectivity (VPR) variation, classify hydrometeors or monitor the  $Z_{DR}$  calibration. Hence, we felt there was a research gap related to the use of QVPs to detect the offset in  $Z_{DR}$  worth to be explored.

- 3. We set "Can the QVPs capture the ZDR offset if constrained to heights/ranges close to the radar and making sure the profiles depict light rain and stratiform events?" as the main research question of the present study, and we believe that we demonstrated that the ZDR offset can be accurately detected if some constraints are applied to classify the QVPs. We consider that we provided a solid foundation of our method in Section 3, but we are aware some ideas were not thoroughly described. We will amend this issue in a revised version of the manuscript taking into account the reviewer's suggestions.
- 4. As mentioned above, we do not consider that our method outperforms the boilerplate practice of detecting the ZDR offset using VPs in light rain. Instead, we reckon we provided an operational "QVP"-wrapper for radars not capable of pointing the antenna at high elevation angles (> 10°). Additionally, we evaluated the proposed QVP-method against VPs built from two radar sites throughout one year of data and an extensive set of disdrometer data located near the radar sites. Although we presented only one case study in the manuscript (Section 4.2.1) for practicality's sake, we did not hand-picked events convenient to our research. Instead, we present a long-term evaluation of our method (as shown in Figures 8-10 in the manuscript) that includes different types of rain events, ranging from light to heavy rain. We consider this evaluation process demonstrates the efficacy of the proposed constraints to filter unsuitable QVPs.
- 5. We want to clarify that we performed a straightforward evaluation of the dry-snow approach to calibrate  $Z_{DR}$  using QVPs. However, several aspects hampered its application on our datasets. First, the dry-snow based method would require a previous hydrometeor classification to detect such types of targets. From our point of view, it is not possible to implement an accurate hydrometeor classifier without having an offset-corrected version of  $Z_{DR}$ . However, we are aware that dry aggregated snow is universally present above the melting layer in stratiform clouds (Ryzhkov et al., 2005). Thus, we analysed hundreds of polarimetric profiles (both VPs and QVPs datasets described in the manuscript) and found that distinctive signatures of dry snow are only clearly visible on the VPs. This agrees with the findings of Ryzhkov et al. (2005) in which signatures of dry snow are clearly visible in QVPs built from higher elevation angles (~  $40^{\circ} - 60^{\circ}$ ). Unfortunately, it is not possible to generate QVPs at such elevation angles using data from the UK Met Office radar network, as the higher tilt for QPE applications is 9°. Conversely, in the QVP data set, we observed values of  $Z_{DR}$  in the rain medium (i.e., below the melting layer (ML) bottom) that contrasted to those observed aloft. Figure 6 of the manuscript exemplify this issue, in which values close to the melting level are different from those observed in liquid media. This is probably due to the beam broadening and non-uniform beam filling (NUBF) effects, expected when the QVPs intercept the ML and regions above at 9° elevations. This is the reason why we cannot use QVPs from these relatively low elevation scans and set dry-snow as the target to derive the ZDR offset.

Moreover, a central claim for this effort seems to follow its **'relative' calibration performance (oversold), esp. for "light rain".** It is unlikely any 'natural' method can genuinely guarantee accuracy better than 0.2-0.3 dB – this has been well-argued by previous authors, including several cited; Prior efforts were rightfully cautious in their claims. Yes, some allowance can also be extended to older studies that are occasionally captives to their moment (i.e., radar technology improves with time  $\rightarrow$  better ability to target lighter rain, etc.). Nevertheless, the intrinsic "light rain" variability is significant and comes in many forms (not limited to):

- Capabilities to provide 'ground truth' (e.g., disdrometers as a poor light rain reference);
- What gets defined as 'light rain' (regional / physical process variability),

- How one identifies these regions with existing radar (Z calibration, etc.), and Location, radar sensitivity/quality, other vertical profile factors (e.g., evaporation, sorting, process) that undermine accuracy claims when averaging over regions.
- 6. We agree with the reviewer on the point that we perform a relative evaluation to the VPs method. Furthermore, the stated accuracy of our method across the manuscript is relative to the VPs and this shall be clearly indicated in the revised version of the manuscript. However, we consider that the long-term comparison between the proposed method and the well-known VP-based ZDR calibration method confirms the validity of our method, as shown in manuscript Figures 7-9. Moreover, we do not only used VPs to assess the performance of our method, but we also validated our results with disdrometer data. We observed good agreement between our results and the disdrometer data, considering the well-known discrepancy when comparing radar data to a fixed point location, as shown in Figures 10-11.
- 7. We agree that the concept of light rain can vary upon several factors, like radar calibration and its geographical conditions. However, we consider that the proposed range (0-20 dBZ) is not that far to what Bechini et al. (2002); Fabry (2015); Yang et al. (2019), amongst others define as light rain (5-35 dBZ), considering that the QVPs represent an average of the PPI scan. Finally, it is worth noting the UK Met Office continuously monitors the quality of the radar reflectivity (Harrison et al., 2012, 2017), hence we consider that this does not add uncertainty into our analysis.

For this reviewer, the authors have not demonstrated they built a better mousetrap. The reviewer understands there is an inevitable overconfidence (aka, marketing) in most manuscripts. However, "relative", not absolute calibration concepts are typically quite conservative, and it should be obvious that selective performance may be better under ideal conditions. The authors' disdrometer image (Figure 5) alludes to some inherent variability in (surface, 'instantaneous') **ZDR properties in "light rain" (aka, dynamic range of intrinsic ZDR** > **0.6 dB**). These depictions are consistent with discussions by Bechini et al., Ryzhkov et al., for what those authors expect from "light rain", or why "light rain" (generic) is less suitable than "dry snow" (see also, specific comment). Select locations (UK) experience different bulk microphysical expectations (e.g., propensity for widespread rainfall, stratocumulus), thus **performances may reflect strong local process** / **natural advantages (e.g., contrast with "light rain" at the peripheries of thunderstorms)**.

- 8. We agree that there is an inherent variability of  $Z_{DR}$  in light rain. As stated in point 5, we cannot use natural targets with no  $Z_{DR}$  variability like dry snow (inherent value close to 0 dB), hence we proposed light rain as target but with thresholds (( $0 < Z_H < 20$ ) that minimise this variability. This range is a compromise to avoid having significant variations on  $Z_{DR}$  but still keeping enough QPVs into the analysis that enable a reliable detection of the  $Z_{DR}$  offset.
- 9. We agree with the reviewer that our disdrometer data may reflect UK local processes. To address this issue, we simulated a wide range of DSDs using the range of parameters described in Bringi and Chandrasekar (2001) expected in real storm events:

$$10^{3} \le N_{w} \le 10^{5} \ [mm^{-1}m^{-3}]$$
$$0.5 \le D_{0} \le 2.5 \ [mm]$$
$$-1 \le \mu \le 5$$
$$R \le 300 \ [mm \ h^{-1}]$$

We randomly generated 10,000 sets of DSD parameters (Nw,  $D_0$  and  $\mu$ ) uniform-distributed within the ranges defined above. Equation 3 from the paper was used to simulate the DSDs, which were used as input to a T-matrix scattering model to compute  $Z_H$  and  $Z_{DR}$ . The scattering simulations were performed using the same assumptions as described in the manuscript: (i) the raindrop shape model from Thurai et al. (2007) (their Eq. 2 for D > 1.5 mm, their Eq. 3 for  $0.7 \le D \le 1.5$  mm, spherical raindrops otherwise); (ii) no canting angle distribution; (iii) maximum diameter for the integration fixed to  $3D_0$ ; (iv) temperature of  $10^{\circ}$  C, radar wavelength of 5.3 cm and elevation angle of  $0^{\circ}$ . The results are shown in Figure 1, which depicts the theoretical variation of  $Z_{DR}$  versus  $Z_H$ . We computed the Zdr bias in light rain and this gives a value of  $\overline{Z}_{DR} = 0.18$ dB for  $Z_H

Revision Figure 1:  $Z_H - Z_{DR}$  dependencies using random values of  $N_w$  /  $D_0$  /  $\mu$

Overall, one takeaway message is that this reviewer does not feel the authors have justified the "QVP" application as a genuine improvement over a generic "average" ZDR monitoring practice, for rain, snow or otherwise. Rather, the reviewer claim may be that "QVPs" in light rain are arguably far worse, given this form of averaging enables mixtures of less suitable profile properties that produce apparently viable "light rain" profiles. Why use a "QVP" process at all? Fundamentally, this is a reduction of information; Many previous studies speak to physical 'profile' issues convolved with "QVPs" and similar averaging, with even the QVP originators shifting to "CVPs" or other targeted averages – For example, ZDR should naturally evolve below the melting layer in response to processes such as sorting, evaporation, break-up, and/or other regime-averaging nuances (within event, or tropical vs midlatitude differences). This all points to why previous studies may have remained cautious in their claims on relative 'light rain' use and uncertainty, but also where QVP-ideas are suboptimal (esp. in rain, below cloud, etc.). The reviewer is questioning the need in using a QVP in these contexts if the QVP cannot be justified as out-performing any number of simpler, targeted ZDR averages of 'light rain' (if one is already thresholding regions loosely on Z, RHV regardless, you've already opened that echo classification bag once one introduced decision-tree thresholding for 'drizzle', etc).

10. We agree that the inherent averaging process in the QVP construction may wash out some key microphysical processes within the precipitation events. However, Ryzhkov et al. (2016) demonstrated the usability of QPVs in radar meteorology, in particular, to monitor the calibration of ZDR. Moreover, we consider that the averaging process to build the QVPs and the proposed constraints to filter profiles not related to light rain is particularly effective in this situation. We proposed several restrictions to identify suitable QVPs that capture the ZDR offset. For instance, our method requires a proper detection of the ML within the QVPs to ensure that the computation of the ZDR offset is reliable. Allabakash et al. (2019); Griffin et al. (2020); Lukach et al. (2021); Sanchez-Rivas and Rico-Ramirez (2021) demonstrated that heights of the ML top and bottom can be accurately estimated using QVPs. We consider that QVPs without ML signatures are filtered by this requirement, thus reducing the uncertainty of using QVPs of polarimetric variables.

**Specific Comment:**

Why do the authors use "light rain" for the "QVP"? Many efforts point to why they avoid light rain (see, Ryzhkov et al, discussions). Unfortunately, the reviewer might have been more amenable to an AMT manuscript that was simply a long-term validation for an existing 'dry snow' QVP concept. That is because most "QVP" concepts and ZDR calibration at higher tilts focus on the properties of lower density, dry aggregate snow as a claimed better-case media. They often note that the spatiotemporal averaging/variability is still a concern, but perhaps less in-cloud and widespread stratiform selective events. Overall, those rationale (e.g., Ryzhkov et al. and subsequent) reflect a somewhat different take on the role of higher tilts and the expected ranges for ZDR media at higher tilts. The current authors use expressions such as:

"The intrinsic value of  $Z_{DR}$  for angles below 90° and collected in light rain is different from zero. Also, it is elevation-dependent, as demonstrated by Bringi and Chandrasekar (2001) and formulated by Ryzhkov et al. (2005) as:"

$$Z_{dr}(\theta) \approx \frac{Z_{dr}(0)}{\left[Z_{dr}^{1/2}(0)\sin^2\theta + \cos^2\theta\right]^2}$$
(8)

11. As stated in point 5, we did carry out a long-term validation of the existing 'dry snow' not only on QVPs but also on VPs. We observed that dry snow is an excellent alternative to calibrate  $Z_{DR}$  using scans taken at vertical incidence, as demonstrated by Ferrone and Berne (2021) or really high elevation angle scans (40°-60°), as shown by Ryzhkov et al. (2005). However, this is not the case for QVPs built from 9° tilts in which targeting the dry snow above the ML exacerbate the NUBF issue; as the range increases, there is a bigger chance of the beam intercepting mixed-phase hydrometeors. This is why we explored the use of light rain in QVPs. We wanted to provide an operational alternative to radars with similar configurations that cannot collect high elevation scans nor birdbath scans.

The reason **Ryzhkov et al. give for higher tilts and dry snow** is seemingly opposite to the current authors' logic – Ryzhkov argues **dry snow has lower natural ZDR variability**, and when these media are viewed from higher tilts (e.g., the eventual multiplier on ZDR in equation (9) would be closer to 0 instead of 1), the dynamic range of potential ZDR variability is low. When the underlying media experiences a wider range of variability, aka, light rain ranges from 0.1 dB to 0.6+ dB at typical trusted Z ranges, etc., this implies added uncertainty for any 'average' reference

frame. These issues are at their most problematic at grazing angles, and possibly not preferable at lower altitudes (given evaporation, other profile physical processing that evolves ZDR below cloud). Thus, it is not immediately preferable (for their concepts) to have:

$$Z_{dr}(\theta = 10^{\circ}) \approx 0.968 Z_{dr}(\theta = 0^{\circ}) \left[ dB \right] \tag{9}$$

e.g., a high coefficient close to 1 is 'bad' for "light rain" in these contexts, b/c the intrinsic ZDR for  $Z \sim 15$ -20 dBz remains in those ranges from 0.1 to 0.6 dBz (aka, author Figure 5); This drives the potential uncertainty against the 'reference' ZDR, and one may be correcting by > 0.3 dB quite often (perhaps this was worse in Oklahoma, where lower, unregulated use of Z caries a wider range of ZDR). High tilt intrinsic property sampling (if available, aka, 'birdbath' at its limit) acts to limit that range of possible ZDR  $\rightarrow$  better chance to accurately pinpoint ZDR. Thus, the authors' statement,

"Hence, ZDR radar measurements collected at elevation angles below 10° are similar to those collected at lower elevation angles and so they do not add additional uncertainty to the offset correction method."

... feels opposite this 'dry snow' rationale. This seems to be a question of whether the authors genuinely believe they can target low-ish variability ZDR 'drizzle' better than low-variability dry snow? This may be regional to the UK, e.g., stratocumulus w/drizzle, but may not seem as reasonable if painted with a US NEXRAD radar lens, as a separate example.

- 12. We agree that methods based on vertical measurements (Gorgucci et al., 1999) or really high tilts (Ryzhkov et al., 2005) are excellent options if such scans are available and we consider that must be used when possible. However, we want to provide an alternative to radars not capable of performing scans at such elevations.
- 13. We agree on the point that dry snow has lower natural  $Z_{DR}$  variability compared to light rain when using high tilts (40°-60°). But this variability increases at lower elevations, and the QVPs are affected by this issue. This is why we restricted the height within the QVPs along with thresholds in  $\rho_{HV}$  in an effort to keep the variability at the minimum. Moreover, we agree with the reviewer that our method may be restricted to QVPs depicting weak, stratiform rain. Still, as mentioned in point 10, it is possible to apply several methods to detect the ML boundaries and stratiform rain events. Thus, we consider that if such conditions are met, our approach yields reliable estimations of the  $Z_{DR}$  offset. We will discuss this further on the revised version of the manuscript.

Even with light rain, these issues are likely worse than presented; For example, this effort has not fully discussed that the disdrometer (Parsivels, etc.) references are poor in light rain R i 1-3 mm/hr. It is unlikely most units capture light rain properties perfectly, esp. with assumptions made for disdrometer processing (a different subset of literature on Parsivel, 2DVD and other light rain comparisons). Dry snow media, similarly, has its own issues with identification, wavelength dependency, complications to "QVP" profiles from non-uniform beam-filling (at C-band, there is potentially intrinsic negative ZDR above the ML owing to non-uniform beam filling!). There is not a quick fix, unfortunately.

14. We performed a new procedure to estimate the  $Z_{DR}$  reference value as described in point 9. We consider that this new approach reduces the uncertainty on using a reference value that may reflect local processes, as it was computed using a range of parameters expected in real storm events. Also, note that the validation provided in Figures 9-11 in the manuscript reflects precipitation events collected throughout one year of data. Hence not only light rain events are analysed, but heavy rain events are included as well.

**References**

- Allabakash, S., Lim, S., and Jang, B. J. (2019). Melting layer detection and characterization based on range height indicator-quasi vertical profiles. *Remote Sensing*, 11(23).
- Bechini, R., Gorgucci, E., Scarchilli, G., and Dietrich, S. (2002). The operational weather radar of Fossalon di Grado (Gorizia, Italy): Accuracy of reflectivity and differential reflectivity measurements. *Meteorology and Atmospheric Physics*, 79(3-4):275–284.
- Bringi, V. N. and Chandrasekar, V. (2001). *Polarimetric Doppler Weather Radar*. Cambridge University Press, Cambridge ; New York.
- Fabry, F. (2015). *Radar meteorology: principles and practice*. Cambridge University Press, Cambridge, United Kingdom.
- Ferrone, A. and Berne, A. (2021). Dynamic differential reflectivity calibration using vertical profiles in rain and snow. *Remote Sensing*, 13(1):1–24.
- Gorgucci, E., Scarchilli, G., and Chandrasekar, V. (1999). A procedure to calibrate multiparameter weather radar using properties of the rain medium. *IEEE Transactions on Geoscience and Remote Sensing*, 37(1 PART 1):269–276.
- Griffin, E. M., Schuur, T. J., and Ryzhkov, A. V. (2018). A polarimetric analysis of ice microphysical processes in snow, using quasi-vertical profiles. *Journal of Applied Meteorology and Climatology*, 57(1):31–50.
- Griffin, E. M., Schuur, T. J., and Ryzhkov, A. V. (2020). A polarimetric radar analysis of ice microphysical processes in melting layers of winter storms using s-band quasi-vertical profiles. *Journal of Applied Meteorology and Climatology*, 59(4):751–767.
- Harrison, D., Norman, K., Darlington, T., Adams, D., Husnoo, N., and Sandford, C. (2017). The evolution of the Met Office radar data quality control and product generation system: RADARNET. *37th Conference on Radar Meteorology*, page 14B.2.
- Harrison, D. L., Norman, K., Pierce, C., and Gaussiat, N. (2012). Radar products for hydrological applications in the UK. *Proceedings of the Institution of Civil Engineers - Water Management*, 165(2):89–103.
- Lukach, M., Dufton, D., Crosier, J., Hampton, J. M., Bennett, L., and Neely III, R. R. (2021). Hydrometeor classification of quasi-vertical profiles of polarimetric radar measurements using a top-down iterative hierarchical clustering method. *Atmospheric Measurement Techniques*, 14(2):1075– 1098.
- Ryzhkov, A. V., Giangrande, S. E., Melnikov, V. M., and Schuur, T. J. (2005). Calibration issues of dualpolarization radar measurements. *Journal of Atmospheric and Oceanic Technology*, 22(8):1138– 1155.
- Ryzhkov, A. V., Zhang, P., Reeves, H., Kumjian, M., Tschallener, T., Trömel, S., and Simmer, C. (2016). Quasi-vertical profiles-A new way to look at polarimetric radar data. *Journal of Atmospheric* and Oceanic Technology, 33(3):551–562.
- Sanchez-Rivas, D. and Rico-Ramirez, M. A. (2021). Detection of the melting level with polarimetric weather radar. Atmospheric Measurement Techniques, 14(4):2873–2890.
- Thurai, M., Huang, G. J., Bringi, V. N., Randeu, W. L., and Schönhuber, M. (2007). Drop Shapes, Model Comparisons, and Calculations of Polarimetric Radar Parameters in Rain. *Journal of Atmospheric and Oceanic Technology*, 24(6):1019–1032.

- Trömel, S., Ryzhkov, A. V., Hickman, B., Mühlbauer, K., and Simmer, C. (2019). Polarimetric Radar Variables in the Layers of Melting and Dendritic Growth at X Band—Implications for a Nowcasting Strategy in Stratiform Rain. *Journal of Applied Meteorology and Climatology*, 58(11):2497– 2522.
- Yang, Z., Liu, P., and Yang, Y. (2019). Convective/Stratiform Precipitation Classification Using Ground-Based Doppler Radar Data Based on the K-Nearest Neighbor Algorithm. *Remote Sensing*, 11(19):2277.

---

## Author Comment (AC2)

**Interactive comment on "Calibration of radar differential reflectivity using quasi-vertical profiles". Response to Anonymous Referee 2**

Daniel Sanchez-Rivas and Miguel Angel Rico-Ramirez

October 9, 2021

We thank the reviewer for the detailed, positive remarks that surely will help us to improve the work. The answers to the questions are given below in blue with the proposed modifications to the paper.

This study proposes an operational method to estimate a systematic bias of radar differential reflectivity (Zdr) using quasi-vertical profiles (QVP). The authors compared the results of the proposed QVP method with those derived from vertical profiles (VP) and disdrometer data for one year period of 2018. They concluded that the new approach is consistent with the traditional method and is operationally applicable.

I think that this study is very important for radar quantitative precipitation estimation (QPE) based on polarimetric variables. However, I see a limitation of this study for an operational application. After reading the manuscript carefully, I found that the QVP method requires a disdrometer-derived Zdr bias for light rain (e.g., 0.18 dB). This is a challenge where there is no disdrometer near radar sites. Additionally, using the disdrometer data in the QVP procedure (e.g., Zdr correction) affects an independent evaluation based on Zdr derived from the disdrometer data (e.g., Fig. 10). My detailed comments are provided below.

We thank the reviewer for raising these important points. Indeed, the method requires a disdrometer-derived Zdr bias in light rain. However, this value can be computed using measured or simulated DSDs. In the paper we used measured DSDs, but the results are the same if simulated DSDs are used (see replies to points 1 and 2 below and Figure 1 in this document). The second point is about using the same disdrometer data set to compute the Zdr bias in light rain and to validate the results. It is fair to say that the validation of the method is performed not only in light rain, but also in moderate and heavy rain. However, to address this issue, we have now simulated a wide range of DSDs to validate the method (see reply to point 2 below).

Major comments:

**1. Title is misleading**

Just looking at the title, I started reading the manuscript with high hope to see how the QVP method can estimate a Zdr bias. However, it turns out that the method needs a reference Zdr value simulated from disdrometer measurements. This is a limitation for the operational estimation for most radar sites, particularly in the United States. I think that the author should include "disdrometer data" in the title.

We believe there is no need to include "disdrometer data" in the title of the manuscript. The proposed method can be applied even if disdrometer observations are not available. We originally computed the Zdr bias in light rain using measured DSDs, but this bias can also be calculated using simulated DSDs. The results show that the simulated value is consistent with the value obtained using measured DSDs (see reply to point 2). Therefore, the proposed  $Z_{DR}$  bias in light rain can be extrapolated to other radar sites.

2. Independent evaluation

Part of evaluation in this study is not independent. The disdrometer data used in the QVP procedure were also used in the evaluation (e.g., Figs. 10 and 11).

To address this issue, we simulated a wide range of DSDs using the range of parameters described in Bringi and Chandrasekar (2001) expected in real storm events:

$$10^{3} \leq N_{w} \leq 10^{5} \ [mm^{-1}m^{-3}]$$

$$0.5 \leq D_{0} \leq 2.5 \ [mm]$$

$$-1 \leq \mu \leq 5$$

$$R \leq 300 \ [mm \ h^{-1}]$$

We randomly generated 10,000 sets of DSD parameters (Nw,  $D_0$  and  $\mu$ ) uniform-distributed within the ranges defined above. Equation 3 from the paper was used to simulate the DSDs, which were used as input to a T-matrix scattering model to compute  $Z_H$  and  $Z_{DR}$ . The scattering simulations were performed using the same assumptions as described in the manuscript: (i) the raindrop shape model from Thurai et al. (2007) (their Eq. 2 for D > 1.5mm, their Eq. 3 for  $0.7 \le D \le 1.5$  mm, spherical raindrops otherwise); (ii) no canting angle distribution; (iii) maximum diameter for the integration fixed to  $3D_0$ ; (iv) temperature of  $10^{\circ}$  C, radar wavelength of 5.3 cm and elevation angle of  $0^{\circ}$ . The results are shown in Figure 1, which depicts the theoretical variation of  $Z_{DR}$  versus  $Z_H$ . We computed the Zdr bias in light rain and this gives a value of  $\overline{Z}_{DR} = 0.18dB$  for  $Z_H

Figure 1:  $Z_H - Z_{DR}$  scatterplot using simulated DSDs.

3. Line 95

Could the author specify the elevation angle of birdbath scans? Based on "averaging azimuthally," the elevation angle is not 90 degrees.

These are scans collected by pointing the antenna vertically (elevation angle of 90deg) while at the same time the antenna rotates around its axis (from 0 to 360 deg in azimuth). We will clarify this in the manuscript.

4. Line 169

Why not a "solid phase?" I think that Zdr for solid phase should be reliable (for VP) if the authors avoid the melting layer (e.g., mixed phase) as seen in Fig. 2 (right).

We agree with the reviewer. Dry snow is an excellent alternative to calibrate  $Z_{DR}$  using scans taken at vertical incidence, as demonstrated by Ferrone and Berne (2021). Furthermore, we explored the  $Z_{DR}$  values above the ML in our VPs dataset and confirmed the usability of dry-snow to compute the  $Z_{DR}$  offset. However, such hydrometeors are not a good option for calibrating  $Z_{DR}$  based on QVPs built from low elevation scans. Targeting areas above the melting layer exacerbate the beam broadening and non-uniform beam filling problems as the range increases. These circumstances complicate using dry snow or other solid phase targets to detect the  $Z_{DR}$  offset on QVPs built from 9° elevation scans. Thus, we selected the use of light rain on both schemes to keep consistency between methods.

5. Figure 2

The lines indicating the ML and ML bottom are different between right and left panels. The individual profile shown in the right panel depicts exactly the same data as in the HTI plot. However, we agree that the lines indicating the ML in the left and right panels do not use the same thickness. We will modify these lines to be consistent in both panels.

6. Line 217

What are "the mean dependencies?"

This refers to the Zdr bias in light rain ( $0 < Z_H < 20 \, dBZ$ ) We will clarify this in a revised version of the manuscript.

7. Line 219

Is the value 0.18 dB supposed to dynamically change depending on different event cases in an operational situation? Otherwise, is this value static?

We proposed this value as the intrinsic  $Z_{DR}$  value expected in light rain at ground level on measured and simulated DSD measurements. Thus, we do not expect this value to vary if the physical process leading to the light rain remains similar.

8. Line 236

Please remove the negative sign in "-0.18 dB." *Noted, thanks for noting this!*

9. Line 239

Why does Zdr offset fluctuate hourly? Is it a mechanical issue? Previous works found hourly variations on the computed  $Z_{DR}$  offset (see Chu et al. (2019); Holleman et al. (2010), for instance). This was not the case in our datasets, where the greatest variations on the  $Z_{DR}$  offset were related to updates on the radar configuration.

10. Figure 4

Please insert a legend for lines with different colors.

Agreed. We will modify this figure accordingly and show all the lines in black (no colour is necessary).

11. Figure 7

While values with VP look consistent, what is the reason of variations with the QVP method in the insets?

Due to the inherent averaging process in the construction of the QVPs, the spatial variation of rain events could lead to QVPs that do not fully represent light rain producing some variability in the estimation of the  $Z_{DR}$  offset. We will discuss the limitations of our approach in more detail in a revised version of the manuscript.

12. Line 293

Please replace "The top row of Figure 10" with "Figure 10(a)." *Noted.*

13. Line 344

Please provide more details about "vague polarimetric signatures." *Noted. We will provide more details on this.*

**References**

Bechini, R., Baldini, L., Cremonini, R., and Gorgucci, E. (2008). Differential reflectivity calibration for operational radars. *Journal of Atmospheric and Oceanic Technology*, 25(9):1542–1555.

- Bringi, V. N. and Chandrasekar, V. (2001). *Polarimetric Doppler Weather Radar*. Cambridge University Press, Cambridge ; New York.
- Bringi, V. N., Thurai, M., Nakagawa, K., Huang, G. J., Kobayashi, T., Adachi, A., Hanado, H., and Sekizawa, S. (2006). Rainfall Estimation from C-Band Polarimetric Radar in Okinawa, Japan: Comparisons with 2D-Video Disdrometer and 400 MHz Wind Profiler. *Journal of the Meteorological Society of Japan*, 84(4):705–724.
- Chu, Z., Liu, W., Zhang, G., Kou, L., and Li, N. (2019). Continuous monitoring of differential reflectivity bias for C-band polarimetric radar using online solar echoes in volume scans. *Remote Sensing*, 11(22).
- Ferrone, A. and Berne, A. (2021). Dynamic differential reflectivity calibration using vertical profiles in rain and snow. *Remote Sensing*, 13(1):1–24.
- Giangrande, S. E. and Ryzhkov, A. V. (2005). Calibration of dual-polarization radar in the presence of partial beam blockage. *Journal of Atmospheric and Oceanic Technology*, 22(8):1156–1166.
- Holleman, I., Huuskonen, A., Kurri, M., and Beekhuis, H. (2010). Operational monitoring of weather radar receiving chain using the sun. *Journal of Atmospheric and Oceanic Technology*, 27(1):159–166.
- Ryzhkov, A. V., Giangrande, S. E., Melnikov, V. M., and Schuur, T. J. (2005). Calibration issues of dualpolarization radar measurements. *Journal of Atmospheric and Oceanic Technology*, 22(8):1138– 1155.
- Thurai, M., Huang, G. J., Bringi, V. N., Randeu, W. L., and Schönhuber, M. (2007). Drop Shapes, Model Comparisons, and Calculations of Polarimetric Radar Parameters in Rain. *Journal of Atmospheric and Oceanic Technology*, 24(6):1019–1032.

---

## Author Response (AR1)

**Authors response on "Calibration of radar differential reflectivity using quasi-vertical profiles"**

Daniel Sanchez-Rivas and Miguel Angel Rico-Ramirez

November 8, 2021

**1 Response to Anonymous Referee 1**

*We thank the reviewer for the insightful review of the manuscript and the interesting feedback that helped us improve our work. The comments were considered for the revised version of the paper. Please note that we made several changes to the manuscript to describe and address some flaws of the proposed method. In the following, we provide below point-by-point answers (in blue) to the comments; the changes refer to the marked-up version of the manuscript.*

**General Comments:**

This manuscript offers a method for monitoring the ZDR offset of a dual-polarization radar using quasi-vertical profiles (QVP). The method is applied on C-band weather radars in light rain media. The authors suggest accuracy to O[0.1 dB], e.g., potentially in-line with 'bird-bath' calibration (natural media). There are two apparent justifications for this publication: **its improvement compared to previous natural media efforts**, and its QVP application towards these ideas.

1. *Please note that we corrected the statements regarding the accuracy of the proposed method throughout the revised version of the manuscript. Also, note that we are not suggesting that our approach is better or should replace the existing calibration methods based on intrinsic values of natural targets. As with any other method, ours faces different advantages and disadvantages, which are now discussed in Lines 465-583.*

The manuscript is not recommended for publication. The study is functional with elements similar to the typical AMT scope, but the reviewer finds **low value in the 'new' concept/application**. The use of intrinsic liquid properties for ZDR monitoring is well known, origins in low angles and selective ZDR averages (i.e., cell peripheries). **This manuscript adds a "QVP"-wrapper aimed now at liquid media, yet lacks the physical underpinning as to why such methods would improve performance over a boilerplate practice to 'average ZDR in light rain'. These "QVP" concepts are evaluated against a modest dataset, but reads to the reviewer as motivated by convenience and applying a poorly-matched "QVP" concept** (hammer looking for a nail?) in a less-behaved condition (light rain) to be 'novel'. Yet, it seems **a straightforward evaluation of an existing snow QVP application (as less original as that seems) may have been far less controversial.** The authors perhaps unintentionally increased their degree of difficulty (at least, to this reviewer), by leaving the reviewer questioning whether simpler, quicker, or (existing dry snow) options for targeted averaging may be equally/more effective.

2. *We consider that the main contribution of our method is that it reviews the capabilities of QVPs built from low elevation scans to compute the $Z_{DR}$ offset. We are aware that there are other methods based on measurements taken in liquid precipitation, like the one proposed by Bechini et al. (2008) based on $Z_{DR}$ averages on the cell peripheries or the one proposed by Gorgucci et al. (1999) based on vertical observations. Although these are great options and should be used where possible, we consider that we have demonstrated that it is possible to exploit the practicality of QVPs to detect the $Z_{DR}$ offset.*

3. *Please note that we added more examples in the manuscript to demonstrate the efficacy of the proposed method; we consider that this address the problem in giving the impression of hand-picking the evaluation events (see Figure 11 for example).*

4. *In this revised version of the manuscript (Lines 492–509), we provide a discussion on why using dry snow as the target to detect the $Z_{DR}$ offset in low elevation QVPs do not yield acceptable results. Furthermore, we now review the limitations of our method in the discussion section (pages 21–26).*

Moreover, a central claim for this effort seems to follow its **'relative' calibration performance (oversold), esp. for "light rain".** It is unlikely any 'natural' method can genuinely guarantee accuracy better than 0.2–0.3 dB – this has been well-argued by previous authors, including several cited; Prior efforts were rightfully cautious in their claims. Yes, some allowance can also be extended to older studies that are occasionally captives to their moment (i.e., radar technology improves with time → better ability to target lighter rain, etc.). Nevertheless, the intrinsic "light rain" variability is significant and comes in many forms (not limited to):

- Capabilities to provide 'ground truth' (e.g., disdrometers as a poor light rain reference);

- What gets defined as 'light rain' (regional / physical process variability),

- How one identifies these regions with existing radar (Z calibration, etc.), and Location, radar sensitivity/quality, other vertical profile factors (e.g., evaporation, sorting, process) that undermine accuracy claims when averaging over regions.

5. *We agree with the reviewer on the point that we performed a relative evaluation against the VPs method. This is now clearly stated in the revised version of the manuscript (see Lines 347–350 for example). However, we consider that the long-term comparison between the proposed method and the disdrometer data confirms the validity of our approach. We observed reasonable agreement between our results and the disdrometer data, considering the well-known discrepancy when comparing radar data to a fixed point location, as shown in Figures 8-11.*

For this reviewer,the authors have not demonstrated they built a better mousetrap. The reviewer understands there is an inevitable overconfidence (aka, marketing) in most manuscripts. However, "relative", not absolute calibration concepts are typically quite conservative, and it should be obvious that selective performance may be better under ideal conditions. The authors' disdrometer image (Figure 5) alludes to some inherent variability in (surface, 'instantaneous') **ZDR properties in "light rain" (aka, dynamic range of intrinsic ZDR > 0.6 dB)**. These depictions are consistent with discussions by Bechini et al., Ryzhkov et al., for what those authors expect from "light rain", or why "light rain" (generic) is less suitable than "dry snow" (see also, specific comment). Select locations (UK) experience different bulk microphysical expectations (e.g., propensity for widespread rainfall, stratocumulus), thus **performances may reflect strong local process / natural advantages (e.g., contrast with "light rain" at the peripheries of thunderstorms)**.

6. *We agree with the reviewer on the point that the inherent variability of $Z_{DR}$ in light rain adds some uncertainty to our method. Hence, we proposed a constraint $(0 < Z_H < 20)$ in an effort to reduce the variability of $Z_{DR}$. This decision was made based on theoretical values of $Z_H - Z_{DR}$ expected in real storm events (see Figure 4(a)). This is now described in Lines 513-523. Moreover, Figures 7-8 show that this constraint produces good results compared to the traditional method based on VPs, in which the $Z_{DR}$ variability is close to 0 dB due to the circular shape of the raindrops when the radar antenna is rotating about the vertical and averaging over full cycles of $360°$ (Gorgucci et al., 1999).*

7. *Please see the reply given in point 4 on why we discarded the use of natural targets with no $Z_{DR}$ variability like dry snow to detect the $Z_{DR}$ offset using low elevation QVPs.*

8. *We consider that the performance of our method is not contingent on UK local processes, but on the presence of stratiform light rain events instead. Yet, in Lines 593-596 we set the absence of stratiform light rain as a limitation of our method.*

Overall, one takeaway message is that this reviewer does not feel the authors have justified the "QVP" application as a genuine improvement over a generic "average" ZDR monitoring practice, for rain, snow or otherwise. Rather, **the reviewer claim may be that "QVPs" in light rain are arguably far worse**, given this form of averaging enables mixtures of less suitable profile properties that produce apparently viable "light rain" profiles. Why use a "QVP" process at all? Fundamentally, this is a reduction of information; Many previous studies speak to physical 'profile' issues convolved with "QVPs" and similar averaging, with even the QVP originators shifting to "CVPs" or other targeted averages – For example, ZDR should naturally evolve below the melting layer in response to processes such as sorting, evaporation, break–up, and/or other regime–averaging nuances (within event, or tropical vs midlatitude differences). This all points to why previous studies may have remained cautious in their claims on relative 'light rain' use and uncertainty, but also where QVP–ideas are suboptimal (esp. in rain, below cloud, etc.). The reviewer is questioning the need in using a QVP in these contexts if the QVP cannot be justified as out–performing any number of simpler, targeted ZDR averages of 'light rain' (if one is already thresholding regions loosely on Z, RHV regardless, you've already opened that echo classification bag once one introduced decision–tree thresholding for 'drizzle', etc).

9. *We agree that the inherent averaging process in the QVP construction may wash out some key microphysical processes within the precipitation events. However, Ryzhkov et al. (2016) demonstrated the usability of QVPs in radar meteorology, and in particular, to monitor the calibration of $Z_{DR}$. We consider that limiting the QVPs until a specific range/height and several other constraints enable the detection of suitable QVPs that capture the $Z_{DR}$ offset. This is somewhat reflected in the relation of VPs/QVPs that meet the "stratiform light rain" criteria and are included in the analysis, as stated in Lines 369-375 and shown in Figures 5 and 6. As expected, the number of valid VPs is larger than the number of valid QVPs. However, we consider that such QVPs are capable of detecting the $Z_{DR}$ offset.*

**Specific Comment:**

Why do the authors use "light rain" for the "QVP"? Many efforts point to why they avoid light rain (see, Ryzhkov et al, discussions). Unfortunately, **the reviewer might have been more amenable to an AMT manuscript that was simply a long-term validation for an existing 'dry snow' QVP concept.** That is because most "QVP" concepts and ZDR calibration at higher tilts focus on the properties of lower density, dry aggregate snow as a claimed better-case media. They often note that the spatiotemporal averaging/variability is still a concern, but perhaps less **in cloud and widespread stratiform selective events**. Overall, those rationale (e.g., Ryzhkov et al. and subsequent) reflect a somewhat different take on the role of higher tilts and the expected ranges for ZDR media at higher tilts. The current authors use expressions such as:

"The intrinsic value of $Z_{DR}$ for angles below 90° and collected in light rain is different from zero. Also, it is elevation-dependent, as demonstrated by Bringi and Chandrasekar (2001) and formulated by Ryzhkov et al. (2005) as:"

$$Z_{dr}(\theta) \approx \frac{Z_{dr}(0)}{\left[Z_{dr}^{1/2}(0)\sin^2\theta + \cos^2\theta\right]^2} \qquad (8)$$

10. *Please note that we explored the use of dry snow as the target to detect the $Z_{DR}$ offset not only using QVPs but also VPs, as stated in this revised version of the manuscript (Lines 492-506). However, we concluded that issues like beam broadening or non-uniform beam filling prevent the use of such targets based on QVPs built from 9° tilt scans. This is why we set light rain as our target to detect the offset, but we also defined constraints that minimise the uncertainty of using measurements taken in liquid precipitation.*

The reason **Ryzhkov et al. give for higher tilts and dry snow** is seemingly opposite to the current authors' logic – Ryzhkov argues **dry snow has lower natural ZDR variability**, and when these media are viewed from higher tilts (e.g., the eventual multiplier on ZDR in equation (9) would be closer to 0 instead of 1), the dynamic range of potential ZDR variability is low. When the underlying media experiences a wider range of variability, aka, light rain ranges from 0.1 dB to 0.6+ dB at typical trusted Z ranges, etc., this implies added uncertainty for any 'average' reference frame. These issues are at their most problematic at grazing angles, and possibly not preferable at lower altitudes (given evaporation, other profile physical processing that evolves ZDR below cloud). Thus, it is not immediately preferable (for their concepts) to have:

$$Z_{dr}(\theta = 10°) \approx 0.968 Z_{dr}(\theta = 0°) \, [dB] \qquad (9)$$

e.g., a high coefficient close to 1 is 'bad' for "light rain" in these contexts, b/c the intrinsic ZDR for $Z \sim$ 15–20 dBz remains in those ranges from 0.1 to 0.6 dBz (aka, author Figure 5); This drives the potential uncertainty against the 'reference' ZDR, and one may be correcting by $> 0.3$ dB quite often (perhaps this was worse in Oklahoma, where lower, unregulated use of Z caries a wider range of ZDR). High tilt intrinsic property sampling (if available, aka, 'birdbath' at its limit) acts to limit that range of possible ZDR $\rightarrow$ better chance to accurately pinpoint ZDR. Thus, the authors' statement,

> "Hence, ZDR radar measurements collected at elevation angles below 10° are similar to those collected at lower elevation angles and so they do not add additional uncertainty to the offset correction method."

. . . feels opposite this 'dry snow' rationale. This seems to be a question of whether the authors genuinely believe they can target low-ish variability ZDR 'drizzle' better than low-variability dry snow? This may be regional to the UK, e.g., stratocumulus w/drizzle, but may not seem as reasonable if painted with a US NEXRAD radar lens, as a separate example.

11. *We consider that the statement "Hence, $Z_{DR}$ radar measurements collected at elevation angles below 10° are similar to those collected at lower elevation angles and so they do not add additional uncertainty to the offset correction method" was vague, so we modified it accordingly (see Lines 253-256). We hope that this correction clarifies that we are aware that dry snow has lower natural $Z_{DR}$ variability compared to light rain when using high tilts ($> 40°$). But this variability increases at lower elevations and the QVPs are affected by this issue, as discussed in point 10. This is why we restricted the height within the QVPs along with thresholds in $\rho_{HV}$ in an effort to keep the variability at the minimum. Thus, we consider that if such conditions are met, our approach yields reliable estimations of the $Z_{DR}$ offset.*

Even with light rain, these issues are likely worse than presented; For example, this effort has not fully discussed that the disdrometer (Parsivels, etc.) references are poor in light rain R ¡ 1-3 mm/hr. It is unlikely most units capture light rain properties perfectly, esp. with assumptions made for disdrometer processing (a different subset of literature on Parsivel, 2DVD and other light rain comparisons). Dry snow media, similarly, has its own issues with identification, wavelength dependency, complications to "QVP" profiles from non-uniform beam-filling (at C-band, there is potentially intrinsic negative ZDR above the ML owing to non-uniform beam filling!). There is not a quick fix, unfortunately.

12. *We performed a new procedure to estimate the $Z_{DR}$ reference value as described Lines 264–277 and shown in Figure 4(a). We consider that this new approach reduces the uncertainty on using a reference value that may reflect UK local processes, as it was computed using a range of parameters expected in real storm events. Also, note that we provide more examples using data collected throughout one year for the validation of our method. This includes not only light rain events, but heavy rain events as well.*

**2 Response to Anonymous Referee 2**

*We thank the reviewer for the detailed, positive remarks that helped us to improve the work. In the following, we address all their point-by-point comments in blue, outlining our response and how we modified the manuscript. The changes refer to the marked-up version of the manuscript.*

This study proposes an operational method to estimate a systematic bias of radar differential reflectivity (Zdr) using quasi-vertical profiles (QVP). The authors compared the results of the proposed QVP method with those derived from vertical profiles (VP) and disdrometer data for one year period of 2018. They concluded that the new approach is consistent with the traditional method and is operationally applicable.

I think that this study is very important for radar quantitative precipitation estimation (QPE) based on polarimetric variables. However, I see a limitation of this study for an operational application. After reading the manuscript carefully, I found that the QVP method requires a disdrometer–derived Zdr bias for light rain (e.g., 0.18 dB). This is a challenge where there is no disdrometer near radar sites. Additionally, using the disdrometer data in the QVP procedure (e.g., Zdr correction) affects an independent evaluation based on Zdr derived from the disdrometer data (e.g., Fig. 10). My detailed comments are provided below.

*We thank the reviewer for raising these important points. Indeed, the method requires a disdrometer–derived $Z_{DR}$ bias in light rain. However, this value can be computed using measured or simulated DSDs. In the paper, we used measured DSDs, but the results are the same if simulated DSDs are used (see replies to points 1 and 2 below and Figure 4 of the revised version of the manuscript). The second point is about using the same disdrometer data set to compute the $Z_{DR}$ bias in light rain and to validate the results. It is fair to say that the validation of the method is performed not only in light rain but also in moderate and heavy rain. However, to address this issue, we have now simulated a wide range of DSDs expected in real storms in order to compute the intrinsic value of $Z_{DR}$ in light rain, and we used the measured DSDs to validate the method (see reply to point 2 below). Finally, note that we modified the manuscript to show different precipitation events throughout one year of data. Moreover, we added a new case study (Figure 11) to illustrate the performance of the proposed method.*

Major comments:

1. Title is misleading

   Just looking at the title, I started reading the manuscript with high hope to see how the QVP method can estimate a Zdr bias. However, it turns out that the method needs a reference Zdr value simulated from disdrometer measurements. This is a limitation for the operational estimation for most radar sites, particularly in the United States. I think that the author should include "disdrometer data" in the title.

   *We believe there is no need to include "disdrometer data" in the title of the manuscript. The proposed method can be applied even if disdrometer observations are not available. We initially computed the $Z_{DR}$ bias in light rain using measured DSDs, but this bias can also be calculated using simulated DSDs. The results show that the simulated value is consistent with the value obtained using measured DSDs (see reply to point 2). Therefore, the proposed $Z_{DR}$ bias in light rain can be extrapolated to other radar sites.*

2. Independent evaluation
   Part of evaluation in this study is not independent. The disdrometer data used in the QVP procedure were also used in the evaluation (e.g., Figs. 10 and 11).

   *To address this issue, we simulated a wide range of DSDs using a range of parameters expected in real storm events, as described in Lines 264-277 and shown in Figure 4(a) of the revised version of the manuscript (page 12). We consider that this can be viewed now as an independent validation of the proposed method.*

3. Zh– Zdr dependence
   There is no Zh– Zdr dependence demonstrated in the manuscript. I think that the authors took simple averages of Zdr values conditioned on Zh values (0–20 dBZ) at each different disdrometer location.

   *The $Z_H$– $Z_{DR}$ dependence is now shown in Figure 4(a) of the revised manuscript, which is also consistent with previous studies (see Bechini et al. (2008); Bringi et al. (2006); Giangrande and Ryzhkov (2005); Ryzhkov et al. (2005) for instance.). See also reply to point 2.*

4. Discussion section
   The discussion section seems to be the summary of this study. Most of the paragraphs are summaries of the results presented in the figures described in the previous sections. I would like to see actual discussions e.g., regarding any challenges or limitations (or sensitive factors) that can affect the accuracy of QVP method. Additionally, there is no "outlook" in the last section.
   *We modified this section and discussed several limitations of the proposed method taking into account the reviewer's suggestions (Lines 465–583).*

Minor comments:

1. Line 4
   Maybe "light rain" instead of "rain?"
   *Corrected in Line 5. Please note that we made slight changes to the abstract to describe the natural targets explored in this work.*

2. Line 4
   Please replace "expected" with "desirable."
   *We consider that "expected" fits better in this context.*

3. Line 95
   Could the author specify the elevation angle of birdbath scans? Based on "averaging azimuthally," the elevation angle is not 90 degrees.

   *These are scans collected by pointing the antenna vertically (elevation angle of $90°$) while at the same time the antenna rotates around its axis (from $0°$ to $360°$ in azimuth). We clarified this issue in Lines 114–115.*

4. Line 169
   Why not a "solid phase?" I think that Zdr for solid phase should be reliable (for VP) if the authors avoid the melting layer (e.g., mixed phase) as seen in Fig. 2 (right).

   *We discussed the advantages and disadvantages of using dry snow and the reasons of using light rain for the $Z_{DR}$ offset detection in Lines 492–509.*

5. Figure 2
   The lines indicating the ML and ML bottom are different between right and left panels.
   *The individual profile shown in the right panel depicts exactly the same data as in the HTI plot. However, we agree that the lines indicating the ML in the left and right panels do not use the same thickness. We modified the figures accordingly, as shown in Figure 5. Please also note that this figure was updated as we consider that this new format enables a better comparison of the methods used to detect the $Z_{DR}$ offset.*

6. Line 217
   What are "the mean dependencies?"
   *This refers to the intrinsic $Z_{DR}$ values expected for a range of $Z_H$ values, in this case, $0 < Z_H < 20$ dBZ. We clarified this in Lines 285–291.*

7. Line 219

   Is the value 0.18 dB supposed to dynamically change depending on different event cases in an operational situation? Otherwise, is this value static?

   *We proposed this value as the intrinsic $Z_{DR}$ value expected in light rain at ground level on simulated DSD measurements (see reply to major comment 2 above). Thus, we do not expect this value to varying if the physical process leading to the light rain remains similar (widespread stratiform precipitation events).*

8. Line 236

   Please remove the negative sign in "–0.18 dB."

   *Fixed in Line 328.*

9. Line 239

   Why does Zdr offset fluctuate hourly? Is it a mechanical issue?

   *Previous works found hourly variations on the computed $Z_{DR}$ offset (see Chu et al. (2019); Holleman et al. (2010), for instance). This was not the case in our datasets, where the greatest variations on the $Z_{DR}$ offset were related to updates on the radar configuration, as shown in Figure 6.*

10. Figure 4

    Please insert a legend for lines with different colors.

    *We modified this figure accordingly and now it shows all the lines in black (no colour is necessary).*

11. Figure 7

    While values with VP look consistent, what is the reason of variations with the QVP method in the insets?

    *Due to the inherent averaging process in the construction of the QVPs, the spatial variation of rain events could lead to QVPs that do not fully represent light rain producing some variability in the estimation of the $Z_{DR}$ offset. We discuss these limitations of our approach in Lines 354–375.*

12. Line 293

    Please replace "The top row of Figure 10" with "Figure 10(a)."

    *Fixed in Line 423.*

13. Line 344

    Please provide more details about "vague polarimetric signatures."

    *Fixed in Lines 556-557.*

**References**

Bechini, R., Baldini, L., Cremonini, R., and Gorgucci, E. (2008). Differential reflectivity calibration for operational radars. *Journal of Atmospheric and Oceanic Technology*, 25(9):1542–1555.

Bringi, V. N. and Chandrasekar, V. (2001). *Polarimetric Doppler Weather Radar*. Cambridge University Press, Cambridge ; New York.

Bringi, V. N., Thurai, M., Nakagawa, K., Huang, G. J., Kobayashi, T., Adachi, A., Hanado, H., and Sekizawa, S. (2006). Rainfall Estimation from C-Band Polarimetric Radar in Okinawa, Japan: Comparisons with 2D-Video Disdrometer and 400 MHz Wind Profiler. *Journal of the Meteorological Society of Japan*, 84(4):705–724.

Chu, Z., Liu, W., Zhang, G., Kou, L., and Li, N. (2019). Continuous monitoring of differential reflectivity bias for C-band polarimetric radar using online solar echoes in volume scans. *Remote Sensing*, 11(22).

Giangrande, S. E. and Ryzhkov, A. V. (2005). Calibration of dual-polarization radar in the presence of partial beam blockage. *Journal of Atmospheric and Oceanic Technology*, 22(8):1156–1166.

Gorgucci, E., Scarchilli, G., and Chandrasekar, V. (1999). A procedure to calibrate multiparameter weather radar using properties of the rain medium. *IEEE Transactions on Geoscience and Remote Sensing*, 37(1 PART 1):269–276.

Holleman, I., Huuskonen, A., Kurri, M., and Beekhuis, H. (2010). Operational monitoring of weather radar receiving chain using the sun. *Journal of Atmospheric and Oceanic Technology*, 27(1):159–166.

Ryzhkov, A. V., Giangrande, S. E., Melnikov, V. M., and Schuur, T. J. (2005). Calibration issues of dual-polarization radar measurements. *Journal of Atmospheric and Oceanic Technology*, 22(8):1138–1155.

Ryzhkov, A. V., Zhang, P., Reeves, H., Kumjian, M., Tschallener, T., Trömel, S., and Simmer, C. (2016). Quasi-vertical profiles–A new way to look at polarimetric radar data. *Journal of Atmospheric and Oceanic Technology*, 33(3):551–562.